# Dissolved Radiotracers and Numerical Modeling in North European Continental Shelf Dispersion Studies (1982–2016): Databases, Methods and Applications

**Pascal Bailly du Bois [1,*], Franck Dumas [2], Claire Voiseux [1], Mehdi Morillon [1], Pierre-Emmanuel Oms [3] and Luc Solier [1]**

[1] Laboratoire de Radioécologie de Cherbourg, IRSN-LRC, rue Max Pol Fouchet B.P. 10, 50130 Cherbourg en Cotentin, France; claire.voiseux@irsn.fr (C.V.); mehdi.morillon@gmail.com (M.M.); luc.solier@irsn.fr (L.S.)

[2] Service Hydrographique et Océanographique de la Marine-SHOM, 29200 Brest, France franck.dumas@shom.fr

[3] Centre d'Etudes et de Valorisation des Algues CEVA, 83 Presqu'île de Pen Lan, 22610 Pleubian, France; omspe@hotmail.fr

**\*** Correspondence: pascal.bailly-du-bois@irsn.fr; Tel.: +33-2-33-01-41-05

**Abstract:** Significant amounts of anthropogenic radionuclides were introduced in ocean waters following nuclear atmospheric tests and development of the nuclear industry. Dispersion of artificial dissolved radionuclides has been extensively measured for decades over the North-European continental shelf. In this area, the radionuclide measurement and release fluxes databases provided here between 1982 and 2016 represent an exceptional opportunity to validate dispersion hydrodynamic models. This work gives accessibility to these data in a comprehensive database. The MARS hydrodynamic model has been applied at different scales to reproduce the measured dispersion in realistic conditions. Specific methods have been developed to obtain qualitative and quantitative results and perform model/measurement comparisons. Model validation concerns short to large scales with dedicated surveys following the dispersion: it was performed within a two- and three-dimensional framework and from minutes and hours following a release up to several years. Results are presented concerning the dispersion of radionuclides in marine systems deduced from standalone measurements, or according to model comparisons. It allows characterizing dispersion over the continental shelf, pathways, transit times, budgets and source terms. This review presents the main approaches developed and types of information derived from studies of artificial radiotracers using observations, hydrodynamic models or a combination of the two, based primarily on the new featured datasets.

**Keywords:** radionuclide; tracer; data collection; antimony 125 ($^{125}$Sb), tritium ($^{3}$H), dispersion; modeling; English Channel; North Sea; Biscay Bay

## 1. Introduction

Significant amounts of anthropogenic radionuclides have been introduced in ocean waters since 1945. The main origins were the fallout from the atmospheric nuclear tests that occurred before 1980, controlled releases from the nuclear industry and the accidental releases resulting from Chernobyl (1986) and Fukushima Dai-ichi (2011) nuclear power plants. Among these sources, in

Europe, the releases from nuclear fuel reprocessing plants from Sellafield and La Hague were the most important.

Oceanographic sampling campaigns made it possible to measure the dispersion of the artificial radionuclides remaining dissolved in seawater over the past 30 years. These works have demonstrated the interest of dissolved radionuclides as tools for oceanography [1–3]. Over the European continental shelf, extensive in-situ measurements have been performed between 1950 and 2000 by English, German, Belgian and French institutes [4,5] that allowed to draw up the general circulation pathways and water masses transit times in the Irish Sea, the North Sea, the English Channel (Figure 1) and the Arctic ocean [6–11]. The Cherbourg Radioecology Laboratory (IRSN-LRC) has contributed to these studies at the scale of the English Channel and the North Sea since 1988 [12–21]. These works have been ongoing since 1994 in the North-East Atlantic waters, the Celtic Sea, Irish Sea and the Bay of Biscay. During the 1990s, it appears that dissolved radionuclide measurements represent exceptional tools to test and validate marine hydrodynamic models applied to represent short to long term dispersion processes.

This work presents a review and an update of results from the main dissolved radionuclides that have been measured extensively by the IRSN-LRC, i.e., $^3$H, $^{125}$Sb, $^{137}$Cs, $^{134}$Cs, $^{106}$Ru and $^{60}$Co. Other radionuclides could be used as oceanographic tracers, such as $^{90}$Sr, $^{99}$Tc, $^{129}$I, $^{238,\ 239+240}$Pu. Radionuclides with a half-life lower than one year has been sparsely detected as $^{110m}$Ag, $^{54}$Mn, $^{58}$Co and $^{131}$I.

Realistic simulations of the dispersion of soluble substances in the marine environment are essential for management of the marine ecosystem. Such simulations were applied to study the fate of chronic or accidental releases into the sea; they may also be used to feed ecological models that encompass exchanges between the different compartments of the environment: seawater, living organisms and sediments. Such tools are particularly relevant in seas that are subject to strong anthropogenic pressures, such as the macro-tidal seas of north-western Europe.

Various methods have been tested for calculating the behavior of water masses, their advection and their dispersion. Models commonly simulate currents at different resolutions (from a tenth of a meter to kilometers), spatial (1 km–1000 km) and temporal coverage (from hour to decades). While these models produce an accurate representation of tidal levels and associated currents, greater requirements are needed to simulate advection of soluble substances over periods longer than the tidal cycle. The models' ability to reproduce dispersion under realistic conditions of release, wind and tide over several days, weeks or years is a sensitive criterion for assessing their reliability.

Validation of hydrodynamic models applied for realistic simulation of mid- to long term dispersion in seawater requires field data of comparable parameters and coverage. The ideal tracer must have a conservative behavior in the water mass; that is to say, neither fixed by the environmental compartments (sediment, living species) nor modified during its stay in seawater and when subsequently diluted. It must be measurable several hundred or thousand kilometers from its input point (meaning even at very low levels of concentration). The discharge conditions and flow must be well known and it must have few properly controlled origins. The radioactive decay is easily accounted by hydrodynamic models.

Some artificial (issued from nuclear industry) radionuclides released by nuclear plants fully meet these specifications if their half-life is long enough compared to the transit-times in the studied area (from weeks to years). Among them, $^{125}$Sb and $^3$H as HTO have proven to be conserved in seawater over years at the scale of European waters. These radionuclides can be measured at very low concentrations, up to the Atlantic seawater background concentrations (concentrations 40,000 times lower than natural radioactivity for gamma-emitters).

Collaboration between Ifremer-DYNECO-PHYSED oceanographic physicians and IRSN-LRC marine radio ecologists contributes to improve the marine hydrodynamic models used [22–29]. Further studies have associated systematically in-situ measurements with model simulations in order to improve the efficiency of measurements to check model's precision and obtain more reliable and versatile models, applicable to all coastal seas of the European continental shelf. Model/measurements comparisons have been performed at the scale of the English Channel and the

North Sea with targeted two-dimensional (2D) residual and instantaneous models [11,22–24]. This was done in the Bay of Biscay [26] and the Pacific [27] with three-dimensional (3D) models.

This study also included high resolution 2D and 3D model/measurements comparisons at short scale close to a release outfall [25].

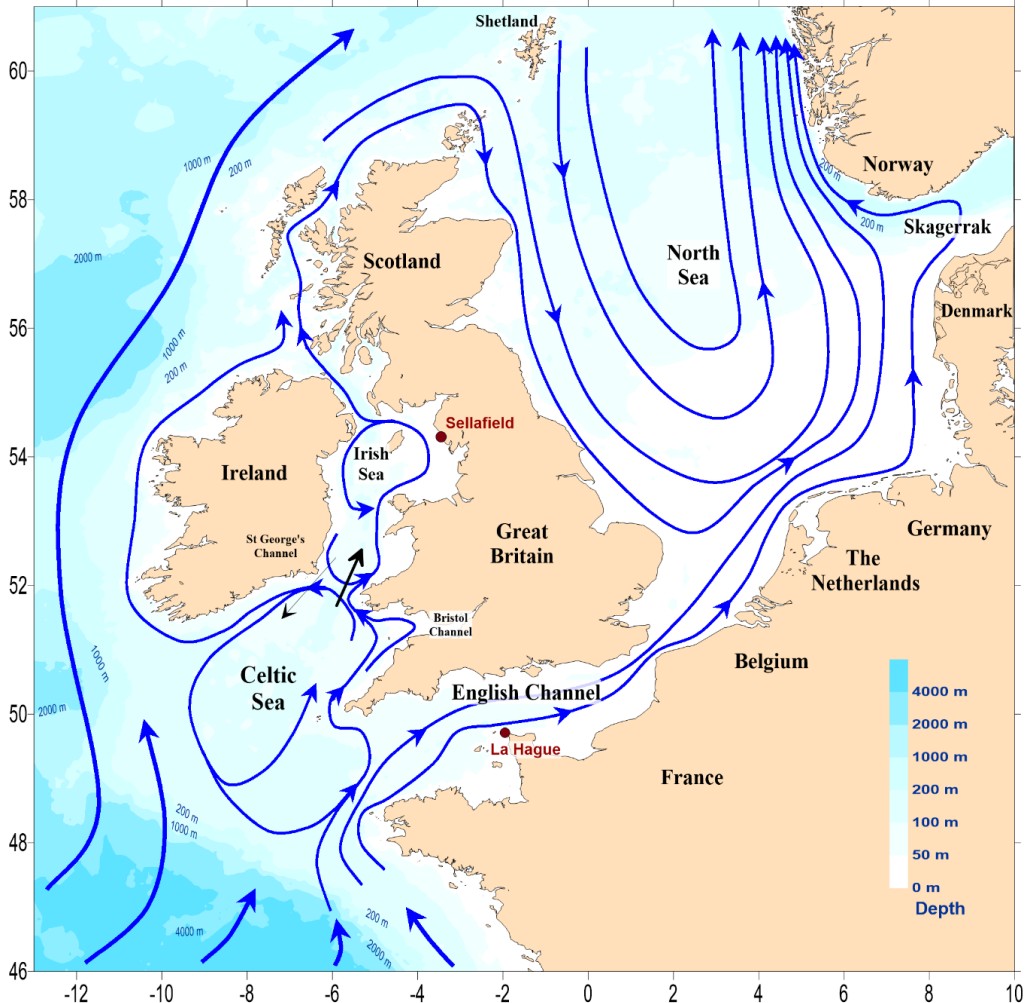

**Figure 1.** General circulation of water masses in the north-west of Europe, adapted from [11].

The purpose of this work is to present a review and an update of field data and methods applicable to validate and improve dispersion models in European macro-tidal seas at all scales. The homogeneous database thus gathered can be used by modelers to test the reliability of their models against appropriate data. It complements the existing databases as the IAEA Marine Information System (MARIS [30]), the World Ocean Database 2013 [31,32], IFREMER–SISMER [33] or BODC [34].

This database has an historical value as releases from nuclear plants have significantly decrease for most gamma emitters (two orders of magnitudes for [125]Sb, [106]Ru, [137]Cs and [60]Co presented here); it represents a huge amount of work (80 oceanographic campaigns) and the dispersion plumes measured before 2000 will be difficult to obtain in the future. Moreover, in a context of a decreasing number of marine radio ecologists, it is valuable to share these data with a larger scientific community. We are convinced that they have not given all the knowledge they could provide.

This work is mainly focused on the dispersion of the Orano recycling plant located at La Hague in the mid English Channel. It represents the main source-term of dissolved radionuclides in this sea. Consequently, its dispersion concerns mostly the English Channel and Southern North Sea.

Nevertheless, other radiotracers source terms have been accounted as Sellafield releases in the Irish Sea or nuclear power plant releases reaching the Bay of Biscay through rivers. It could represent a metrological challenge, but tracers exist to extent the work elsewhere in the world.

This work addresses the following successive aspects: the background section describes successively the radionuclides measured, sampling and measurement methods; the measurement and release database achieved; a short description of the models used and the different model/measurement comparisons performed.

The application section focuses on the main features that could be retained from the point of view of radiotracers, hydrodynamic models and methods, from the short scale in the vicinity of an outfall, to the large scale in the English Channel, North Sea, the Biscay Bay and in the Pacific (Fukushima accident). The last section presents perspectives of applications in other areas or oceanographic domains.

## 2. Background

### 2.1. Dissolved Radionuclides as Oceanographic Tracers

To study the dispersion process over short to long periods or carry out and interpret repeated measurements for varying conditions of release or forcing, it is necessary to use tracers which fulfill the following characteristics:

1.  It must originate from one or a small number of clearly identified release points;
2.  The release conditions must be precisely known (time, fluxes);
3.  Labeling in seawater must be significant, in particular in relation to the pre-existing background level. Labeling concerns seawaters where a significant concentration of radionuclide could be measured.
4.  The tracer must be soluble and not fix onto living organisms or sediments over time (i.e., the stable element is conservative in seawater);
5.  It must be possible to measure the tracer after dilution in the sea over hours, weeks, months or years after its release.

Natural tracers, such as copper, iron, nutrients, cannot generally be used because of the multiplicity of their source terms and the complexity of the phenomena governing their production and fate in the marine environment.

Even if the release conditions are known for artificial tracers, there are often many release points for each one. Such tracers are often involved in geochemical and biogeochemical processes, and therefore their conservative behavior in the marine environment are not guaranteed as shown in Section 3.2.4. for $^{60}$Co and $^{106}$Ru.

Radioactive tracers generally meet criteria 1 and 2. As regards criteria 3, 4 and 5, some radionuclides exhibit long term conservative behaviors in seawater, such as $^{125}$Sb, $^{99}$Tc and $^{3}$H (tritium) [35], and to a lesser extent $^{90}$Sr, $^{137}$Cs and $^{134}$Cs. Between 1970 and 1995 extensive measurements of these radionuclides were carried in the seas of north-western Europe [5,6,11,14–21,24,36]. Reductions in fluxes released during the period from 1980 to 2000 have led to significant decreases in concentrations in the marine system. Out of the radionuclides mentioned above, only tritium released from nuclear fuel re-processing plants has not undergone a reduction since 1980 (Figure 2); nevertheless, it fully satisfies the five criteria, having a radioactive half-life of 12.4 year.

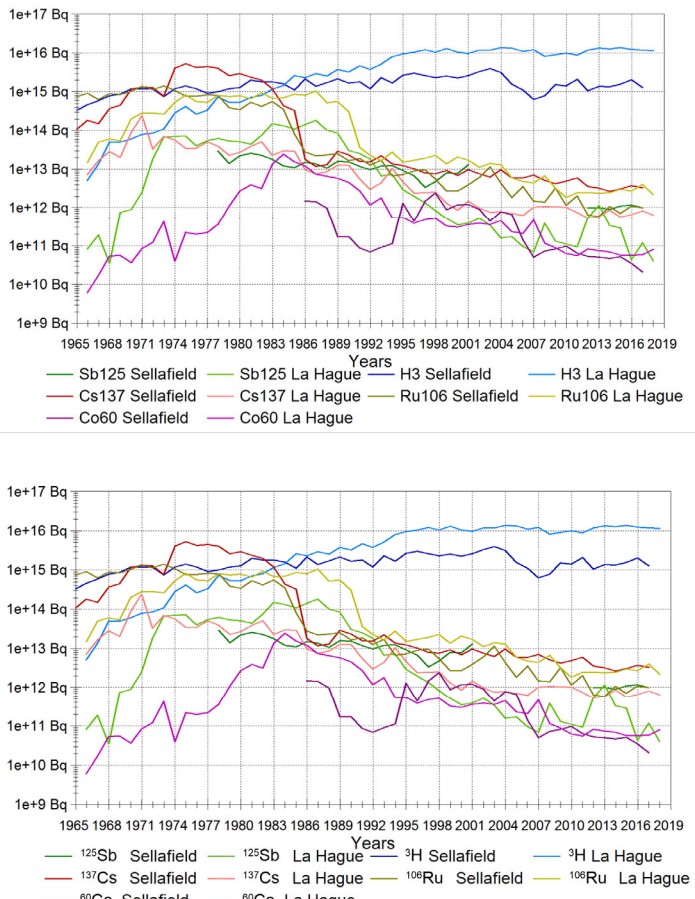

**Figure 2.** Annual liquid releases from Sellafield and La Hague nuclear reprocessing plants for $^{125}$Sb, $^{3}$H, $^{137}$Cs, $^{106}$Ru and $^{60}$Co.

An additional interest of artificial radionuclides as environmental tools is that there is no risk of samples contamination during in-board sampling process, which is generally not the case for other chemicals that require clean conditions to prevent pollutions.

The radionuclides measured in this study that have the most conservative behavior in seawater are mainly antimony 125 ($^{125}$Sb) and tritium ($^{3}$H) in the form of tritiated water HTO (Table 1). The radioactive emission determines the technic that could be applied for each radionuclide.

**Table 1.** Characteristics of the radionuclides used in this work.

| Radionuclide | Tritium $^{3}$H | Cesium 137 $^{137}$Cs | Cesium 134 $^{134}$Cs | Antimony 125 $^{125}$Sb | Ruthenium 106 $^{106}$Ru | Cobalt 60 $^{60}$Co |
|---|---|---|---|---|---|---|
| Radioactive decay (year) | 12.3 | 30.2 | 2.1 | 2.8 | 1 | 5.3 |
| Principal radioactive emission | β | γ | γ | γ | γ | γ |
| Conservative behavior * | 100% | 83%–86% | 83%–86% | 98% | 19%–26% | 8%–14% |

*: Percentage of radionuclide quantities remaining dissolved in seawater at the scale of the English Channel (from [20]).

### 2.1.1. Gamma Emitters

Gamma emitter's ($^{137}$Cs, $^{134}$Cs, $^{106}$Ru and $^{60}$Co in Table 1) concern radionuclides which the main radioactive emission is a high energy γ photon. Thus, they can be detected all together by their

specific energy emission without radiochemistry for the isolation of each element. Specific methods adapted to the dilution conditions observed in the marine environment and the on-board treatment of samples was developed in order to measure artificial activities of the order of 0.3 Bq·m⁻³ [37,38]. This is in contrast with the natural radioactivity of seawater, which is of the order of 12,000 Bq·m⁻³ (mainly $^{40}$K).

This work focuses mainly on $^{125}$Sb and $^{3}$H which are conservative in seawater; nevertheless, the database provides measurements of the radionuclides, which are measured together ($^{106}$Ru, $^{137}$Cs, $^{134}$Cs and $^{60}$Co). They are useful to investigate other process as exchanges between seawater and sediments. Methods to measure $^{125}$Sb, $^{106}$Ru, $^{137}$Cs, $^{134}$Cs and $^{60}$Co have been described in length in [37,38]. Analyses were carried out at the French Navy-Groupe d'études atomiques in Cherbourg (GEA), and at the IRSN-LRC laboratory in Cherbourg.

### 2.1.2. Tritium

$^{3}$H is present in all nuclear plants' liquid releases. The two main sources of tritium are the nuclear fuel reprocessing plants at Sellafield into the Irish Sea that have been active since 1952, and from La Hague into the English Channel since 1966. Liquid releases from each nuclear power plant are two orders of magnitude lower than the fluxes issued from reprocessing plants. Nevertheless, these releases must also be accounted for all along the European coasts and rivers, particularly away from the reprocessing plants influences.

Due to the small amount of seawater required for direct measurement by liquid scintillation (8 mL) a large amount of samples can be processed that can be used to have a high frequency 2D picture or even a 3D picture of the tritium dispersion at a short scale close to the La Hague outfall (Sections 2.4.1. and 2.4.2. [26,39]). The power plants labeling could be used in areas out of theses influences, with corresponding levels lower than 1 Bq·L⁻¹, in the Bay of Biscay [26] or in the Mediterranean Sea.

In the releases performed by nuclear reprocessing plants and French nuclear power plants, tritium is the form of the tritiated water molecule HTO; thus, it has a strict conservative behavior in seawater. This is not always the case: for example, the releases performed by the Amersham plant in Cardiff in the Bristol Channel concern organic compounds tagged with tritium. Organically bounded tritium (OBT) has a specific behavior in marine environment with strong assimilation by living species [40]. Due to their locations (mainly the Bristol Channel and Rhone River), these kinds of releases do not influence the results presented here.

The most used method to measure tritium is liquid scintillation, which allows up to 10,000 samples per year for one analyzer. With a detection limit around 1 Bq·L⁻¹, this method could only be relevant within the plume of reprocessing plant releases (eastern English Channel, Irish Sea, rivers). Extra suited methods exists for tackling lower levels of concentration, which requires sampling one liter of seawater. Low level $^{3}$H measurements were performed with two methods:

(i)　　Electrolytic enrichment of water samples [41]. The detection limit reached 0.01 Bq·L⁻¹.
(ii)　　$^{3}$He regrowth and measurement by mass spectrometry [42]. The detection limit could reach 0.001 Bq·L⁻¹.

### 2.2. Database

### 2.2.1. Radionuclide Measurements in Seawater

The measurement database provided includes all oceanographic campaigns performed by IRSN-LRC (CEA-LRM before 2002) between 1982 and 2016 (Table A1). Data are provided for $^{3}$H, $^{125}$Sb, $^{106}$Ru, $^{137}$Cs, $^{134}$Cs and $^{60}$Co. The database concerns 80 oceanographic campaigns and totals 39,642 sampling locations at sea. Data concerning the coastal monitoring station the closest to the La Hague outfall were added (744 measurements between 1984 and 2018).

Table A1 in Appendix A lists the different campaigns, the number of measurements and maximum concentrations measured. Figure 3 shows the locations of all samples obtained. The

measurement database is available in [43] as a supporting material for this work. Part of these data was already given available in [5] (2010 [137]Cs measurements), in [26,44] (14,494 tritium measurements), and sparsely in previous publications. The new database encompasses 47,387 measurements; more than 60% of these data were previously unpublished.

As said in the introduction, existing databases include measurements of dissolved radioactivity in the European seas. The most important are the IAEA Marine Information System (MARIS [30]), the World Ocean Database 2013 [31,32], the IFREMER–SISMER [33] or the BODC [34]. The data provided here complete them in different ways.

[137]Cs and [134]Cs are commonly measured in oceanographic studies, but [125]Sb, [106]Ru and [60]Co are sparsely detected. [3]H was used as a tracer in open oceans but to a lesser extent in coastal waters.

Due to their conservative behavior in seawater, [125]Sb and [3]H are choice tracers to track the dispersion of industrial releases.

The data collection was designed to appraise the dispersion of French nuclear plants releases from short to large spatial and temporal scales through repeated oceanographic campaigns since 1982. The La Hague controlled radioactive releases in seawater are the more important in Europe; the corresponding radiotracer measurement database allows a complete case study of its dispersion in marine systems.

To our knowledge, no other in-situ measurement database allows to follow individual radionuclides releases in the marine environment at short scale.

The provided database is homogeneous with the release data; it could be included further in other existing databases.

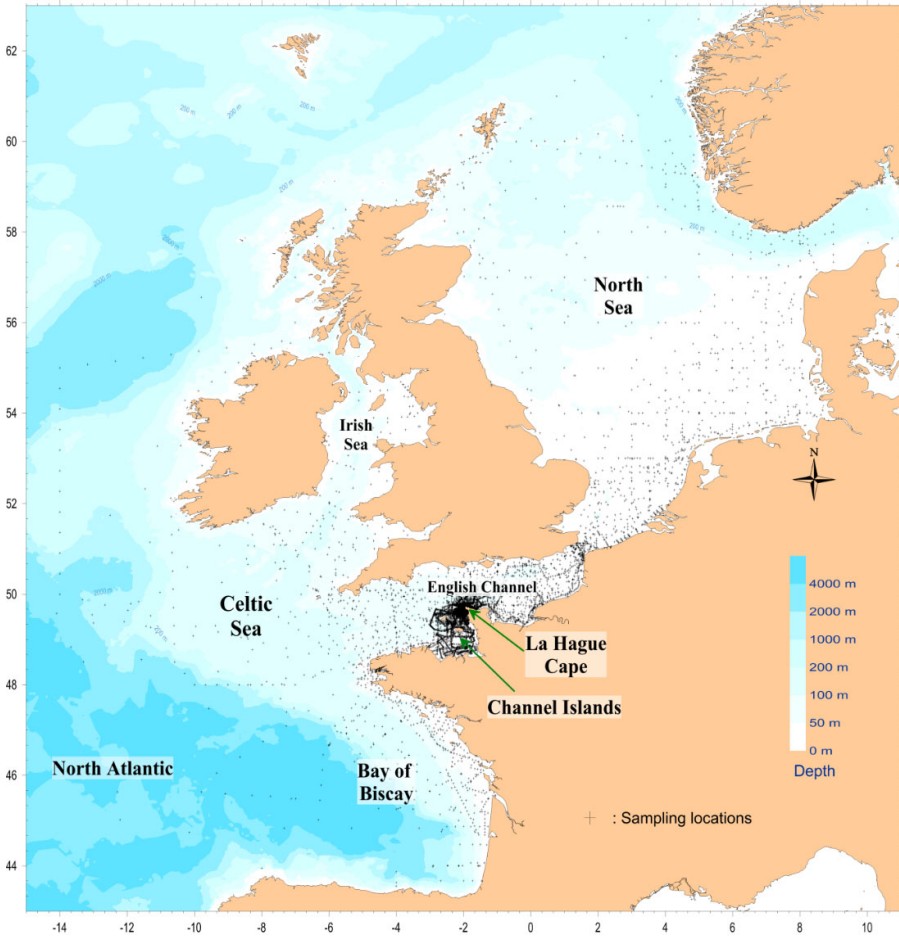

**Figure 3.** Location of radionuclide measurements samples obtained between 1982 and 2016.

2.2.2. Radioactive releases

Releases in seawater from La Hague reprocessing plant was transmitted by Orano company for each individual release since 1982 up to 2018. Quantity released, date-time of the beginning and end of each release are given. Prior to 1982, only annual releases were available. Part of these data was already given available in [26,45] (7840 tritium releases).

The new database encompasses 22,183 individual releases for tritium, $^{137}$Cs, $^{134}$Cs, $^{125}$Sb, $^{106}$Ru and $^{60}$Co (110,915 values). In total, 93% of these data are unpublished as part of a comprehensive accessible database until now.

The Sellafield reprocessing plant annual release data have been published by [46] and in MAFF, RIFE and Sellafield Ltd. reports [47,48]. $^{125}$Sb data have been available only since 1978 with an information gap between 2001 and 2012.

Annual releases from British and French nuclear power plants are issued from annual reports from MAFF, RIFE and EDF reports [47,48].

The releases database is available in [49] as a supplementary materials for this work. Figure 2 presents the annual fluxes released by Sellafield and La Hague reprocessing plants for $^{125}$Sb, $^{137}$Cs, $^{106}$Ru and $^{60}$Co. It shows that the main release period for $^{125}$Sb, $^{3}$H, $^{137}$Cs, $^{106}$Ru and $^{60}$Co occurs before 1992, with a maximum for $^{137}$Cs between 1969 and 1983 at Sellafield (1–5 PBq·y$^{-1}$), and for $^{106}$Ru between 1969 and 1991 at La Hague (close to 1 PBq·y$^{-1}$). Tritium releases (in blue) show a more homogeneous temporal evolution, with fluxes in the same magnitude as Sellafield since 1969 (1–3 PBq·y$^{-1}$) and since 1993 from La Hague (around 10 PBq·y$^{-1}$). It depends mainly from the quantities of nuclear fuel processed by the plants.

Existing datasets concern mainly annual releases from selected nuclear plants. This work provides a compilation of all the release data in one accessible database for French and British nuclear plants in European seas. The knowledge of each individual release is essential to perform precise comparisons between measurements and hydrodynamic model simulations and evaluate the dispersion in the vicinity of an outfall. The detail of the La Hague controlled radioactive releases represent an unparalleled dataset in this perspective.

*2.3. Hydrodynamic Models*

The different models applied are based on the Model for Applications at Regional Scale (MARS) developed by the French Ifremer institute since 1987 [50]. This model was built under various assumptions presented below: using the non-stationary Saint-Venant equations (i.e., 2D); the primitive 3D equations; the "Lagrangian barycentric" method in 2D to filter out the tidal signal.

2.3.1. 2D Modeling

Numerous modeling studies [20,22–25] have demonstrated that models using two-dimensional horizontal approximation (i.e., shallow-water equations) are able to simulate a satisfactory dissolved-substance transport in the non-stratified area. These equations were solved using the finite-difference MARS2D model [50].

In the largest domain the baroclinic effects can be neglected (so that 2D models can relevantly be used) as long as they do not influence that much the targeted area. Naturally, they play a major role over the shelf of the Bay of Biscay or next to the mouth of large estuaries (Loire, Gironde, Rhine etc.). In the eastern English Channel and southern North Sea, away from rivers, the ocean is rarely thermally stratified nor stratified in terms of salinity and temperature; this is due to the strong tidally induced mixing.

2.3.2. 3D Modeling

The three-dimensional MARS3D model is described in detail in [50]. The model uses a 3D finite difference scheme, applying the Boussinesq approximation and hydrostaticity to resolve primitive equations. The model involves a nesting strategy (example Figure 4), starting from a broad region

covering the entire North-West European continental shelf (with a 5.6 km grid resolution) down to a detailed domain covering a few tens of km (with a 5–100 m resolution).

The bathymetry at the grid nodes of the different models is estimated with the method described in [51] from various data sources [25,52].

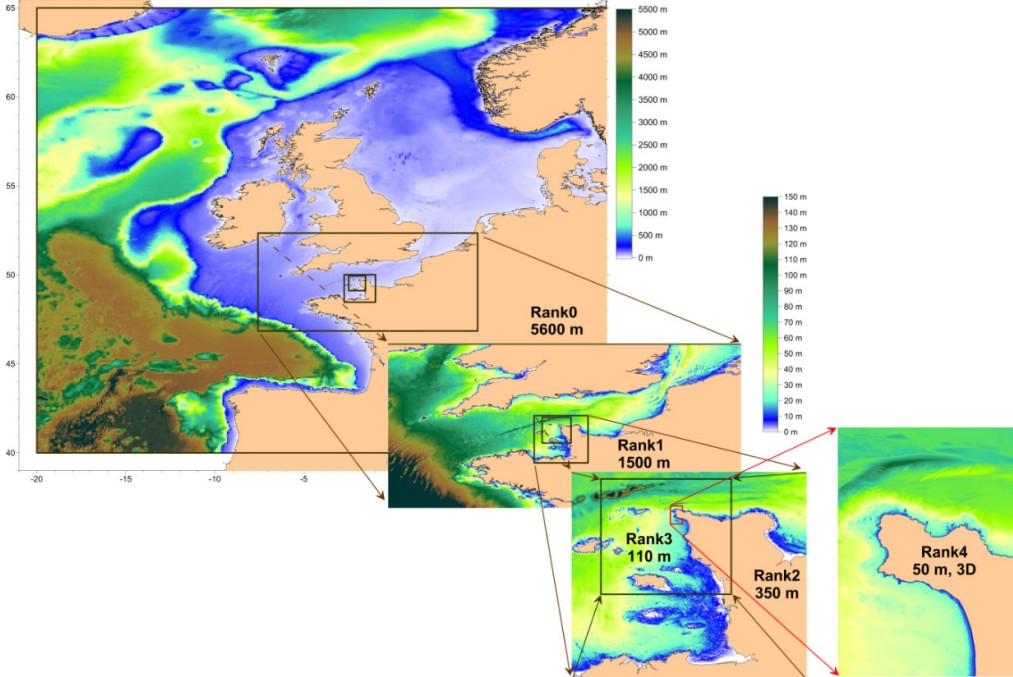

**Figure 4.** Example of model nesting for the La Hague Cape area, Ranks 0–2 are two-dimensional (2D), Rank3 is 2D or three-dimensional (3D), Rank4 is 3D, adapted from [26].

### 2.3.3. 2D "Lagrangian Barycentric" Method

A tidal residual model was designed to reproduce transport, dilution and decay phenomena over long time scales (ranging from a week to several years) and extensive spatial scales (from 30 to 1000 km). It is based on the hypothesis that the water column is homogeneous and that barotropic phenomena will prevail over dynamic baroclinic phenomena. This model has been extensively described and applied in [18,20,25,52–54]. The interest of applying "Lagrangian barycentric" currents is to investigate the residual circulation occurring at long time scales (more than a tidal cycle of 12.42 h), and to simulate dispersion very quickly at the scale of the English Channel and the North Sea (computation time about 1000 times lower than with a similar 2D model).

### 2.4. Sampling Close to an Outfall

### 2.4.1. Model Assisted Sampling

In the vicinity of a punctual source-term (represented by the location of the end of the release pipe), the sampling strategy must be adapted to catch the dispersion of a rapidly moving narrow plume exhibiting short scale features.

During the first hours following release, it is necessary to have good knowledge of the time schedule of the release and where the plume will be located close to the outfall. The dynamics of the currents and duration of releases close to the La Hague Cape impose precise positioning in time (less than 15 min) and space (100 m), to ensure that the sampling sections of the ship's track encompass the plume area (Figure 5 [25]). The releases schedules were transmitted to the vessel by the nuclear plant unit (Service de Protection Radiologique-SPR) of the ORANO company. The MARS2D model (see Section 2.3.1., rank 3 in Figure 4) is used on-board to get some forecasts of the dispersion ahead

of the release and allow precise positioning of the vessel during the hours following the start of the release.

The procedure made possible to sample at high frequency (30 s) an average of 300 surface stations to follow the horizontal dispersion of a given release (see for example the figure in Section 3.1.3.). Varying hydrodynamic and meteorological conditions were investigated.

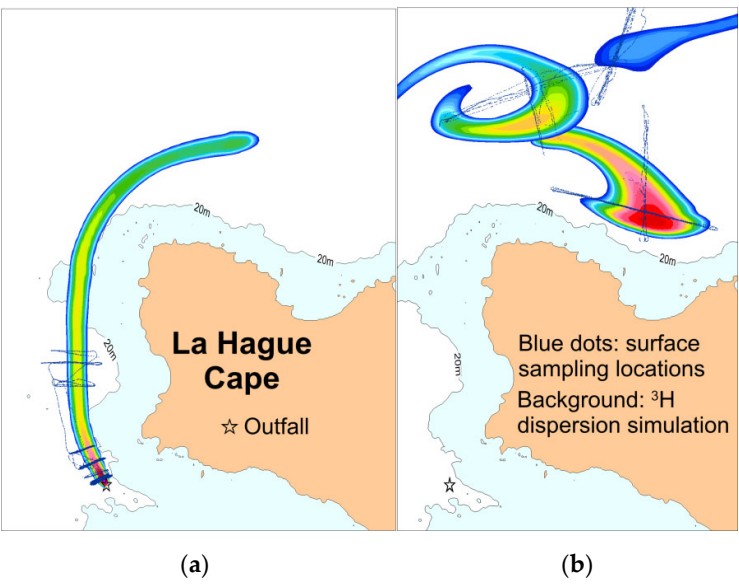

(**a**)　　　　　　　　　　　　　　　　　　　(**b**)

**Figure 5.** Sampling locations during the DISVER 2011 campaign (13,400 in-depth samples between 5 and 11 April 2011) (**a**) 0–2 hours after a release (**b**) 3–8 hours after a release.

2.4.2. High Frequency In-Depth Sampling

Sampling at depth during oceanographic campaigns requires usually bottles deployed from the surface down to the sea floor on a manual or monitored sampling system, such as a CTD rosette. This requires the ship stops at stations during sampling event (i.e., move the sampling system downward, close bottles and move upward at around 1 m·s⁻¹ speed). Overall, it takes about 15–30 min for each station in shallow waters (20–50 m). In the working area, for security reasons the ship cannot be stopped close to the coast or in areas with strong currents. Furthermore, the plume identification requires much higher sampling rates. In the context of the DISVER experiment (Section 3.1.2. [55]) for studying vertical dispersion close to an outfall, we aim to validate three-dimensional hydrodynamic models by using in-situ in-depth radiotracer measurements; this objective requires high-frequency sampling (one every 30 s) at 10 depth levels simultaneously (thus giving 1800 samples per hour).

The area off the Cap de La Hague represents one of the most difficult areas to investigate for such studies, since it combines strong currents (up to 5 m·s⁻¹), proximity of the coast (less than one km) and a complex topography with many rocky shoals and deeps varying from 25 to 90 m over a few kilometers. For operational and security reasons, the ship must remain under way normally at speeds from 1 to 5 m·s⁻¹ (10 knots) while sampling. A devoted system was developed to perform safe sampling in such rough environment [39].

It comprises three main components:

- A sampling line designed to sample 10 depths simultaneously down to 65 m;
- A deep towed depressor (known as Dynalest), which maintains the line at depth close to the seabed;
- An automatic high-frequency sampler with volume, flux and depth control to get 1800 samples per hour.

During sampling, the ship was operating at speeds ranging in 0.5–5 m·s$^{-1}$ with frequent U-turns. Around 13,000 samples could be collected during a four days survey. Vertical slice pictures of the dispersion plume were thus obtained each 5–10 min (100–200 measurements each slice).

The combined system as configured is adapted for in-depth sampling when at least one of these conditions is required: high frequency sampling, proximity of reefs or coasts, strong currents, several depths simultaneous sampling.

*2.5. Methods for Model/Measurement Comparisons*

2.5.1. Comparison with Individual Measurements

Comparisons between measured and simulated radionuclides concentrations are performed at the same x, y, z and t (longitude, latitude, depth and time). Different statistics may be computed:

Correlation coefficient R:

$$R = \frac{\sum_{n=1}^{N}(Meas._n - \overline{Meas.})(Sim._n - \overline{Sim.})}{\sqrt{\sum_{n=1}^{N}(Meas._n - \overline{Meas.})^2}\sqrt{\sum_{n=1}^{N}(Sim._n - \overline{Sim.})^2}} \tag{1}$$

95th percentile of the ratio (PR) between measured and computed concentrations:

$$PR = Max\left[\frac{Meas._n}{Sim._n}, \frac{Sim._n}{Meas._n}\right] \tag{2}$$

Absolute percentage error APE:

$$APE = 100\%\frac{|(Meas._n - Sim._n)|}{Meas._n} \tag{3}$$

Mean absolute percentage error MAPE:

$$MAPE = \frac{100\%}{N}\sum_{n=1}^{N}\frac{|(Meas._n - Sim._n)|}{Meas._n} \tag{4}$$

where:

Meas.$_n$: Measured tritium concentration of the n sample (Bq·m$^{-3}$);
$\overline{Meas.}$: Mean measured tritium concentration (Bq·m$^{-3}$);
Sim.$_n$: Simulated tritium concentration of the n sample (Bq·m$^{-3}$);
$\overline{Sim.}$: Mean simulated tritium concentration (Bq·m$^{-3}$);
N:  Number of samples.

A histogram of the absolute percentage errors could be drawn in order to assess the risk of the model to misestimate the real concentration. Examples of uses of these criteria are present in [24,26,56].

In the vicinity of an outfall, in case of highly variable plume distribution a small change of location (50 m) may result in large concentration variations. Other parameters must then be applied to check for the model reliability [25].

Additional comparison criteria suited for near outfall field.

- Maximum concentration in the plume;
- Mean concentration in the plume;
- Width of plume intersected;
- Distance between maximal measured and simulated plume positions;
- Plume width discrepancy;
- Average and maximum concentration discrepancies;
- Dilution rate discrepancy.

To compare several campaigns with variable source terms, measured concentrations could be normalized by the cumulated released flux corresponding to the plume targeted. They become dilution coefficients (DC) or dilution factors with:

$$DC = \frac{Bq \cdot L^{-1}{}_M}{Bq \cdot L^{-1}{}_R} \tag{5}$$

$Bq \cdot L^{-1}{}_M$: measured concentrations
$Bq \cdot L^{-1}{}_R$: released flux

This dilution coefficient was applied for model/measurement comparisons at a short scale in the vicinity of an outfall, when individual release plumes could be distinguished [25]. At larger scales, individual releases are mixed together in seawater. A more representative approach is to compare the measured concentration ($Bq \cdot m^{-3}$) with the mean released flux in $Bq \cdot s^{-1}$. Such dilution coefficients (concentrations corresponding to a given released flux) were applied at the scale of the whole English Channel and the whole North Sea (see Section 3.2.3., [20,24]).

For 3D model/measurement comparisons, the methods described previously could be applied by accounting for the water mass stratification of the radionuclide concentrations. In case of studies in the near outfall field, particular attention should be given to obtain the same reference of x, y, z and t between measurements and simulations: an error of minutes or hundred meters could result in a concentration variation larger than an order of magnitude.

2.5.2. Radionuclides Inventories

When campaigns had a sufficient extent and sampling rate, localized radioactivity measurements were interpolated over the whole studied area at the nodes of a regular grid. A well-suited interpolation method is "kriging" [57]. This allows the visualization and comparison of the distribution plumes measured and simulated (see Section 3.2.1.).

If the radionuclide distribution in seawater is not stratified (no variation from surface to sea bottom), the total inventory of radionuclides present could be simply calculated by the multiplication of the surface concentration C by the volume of seawater of each mesh (C × dx × dy × depth). In case of stratification, in-depth measurements must be performed in order to associate the right concentration to each level of the water column [26].

Quantities of measured radionuclides can be straightforwardly compared to the simulated ones at different scales [18,20,24,26,27] or compared to the known releases in order to exhibit transit times or conservative behavior [18]. Inverse calculation has also been applied to estimate the source term (e.g., in the Fukushima case) [58].

*2.6. Scales for Model/Measurement Comparisons*

Different time and space scales are presented in this work; the distinction of the different scales is based on the dispersion characteristics in a macro-tidal context. Super tidal scale (minutes to hours, 100 m–10 km) concerns dispersion close to an outfall when the labeled plume is not vertically averaged. The tidal scale (hours to days, 1–30 km) concerns the dispersion in the vicinity of an outfall when the labeled plumes associated to each specific release could be distinguished by surface measurements. Large scale (from week to years, 30–1000 km) concerns the dispersion when individual plumes are mixed together. This is discussed further in Section 4.3.

**3. Applications**

The obtained database [43] contains data from 1982 up to 2016, with all measurements acquired by the IRSN-LRC laboratory during oceanographic campaigns. It concerns mainly $^{125}$Sb, $^{3}$H, $^{106}$Ru, $^{137}$Cs, $^{134}$Cs and $^{60}$Co.

Such data would have been meaningless without the corresponding known releases that explain the observed labeling (cf. 2.2.2. [49]).

*3.1. La Hague Cape, Short Scale/High Resolution Studies*

3.1.1. The La Hague Cape Main Characteristics for Dissolved Radionuclide Dispersion

The North Cotentin includes several nuclear facilities, such as the nuclear power plant at Flamanville Cape, the building of nuclear submarines at Cherbourg and the ANDRA low-level waste deposit at La Hague. The most important in term of liquid releases at sea is the Orano reprocessing plant, of which the outfall is located next to the La Hague Cape (Figure 5). The La Hague Cape forms a physical boundary between the Normandy–Brittany Gulf in the south-west, and the mid-English Channel towards the east. Because of the coastal morphology, the tidal wave coming from the Atlantic is partly blocked in the west-facing bay formed by the Normandy-Brittany Gulf. This embayment is characterized by very large tidal ranges (reaching more than 14 m near the Mont St Michel during spring tides). The Cape de la Hague works as a bottleneck for the water masses involved during the emptying and filling of this bay twice a day. This explains why the tidal currents close to the cape are among the strongest in Europe (they can reach 5 m·s$^{-1}$ during spring tide), with highly variable tidal range around the cape (from 11 m in the south of the Cape down to 6 m in the north [59]). This area is also characterized by diverse topography exhibiting pronounced gradients (depths from 20 to 100 m), many islands, numerous bays and shallow coves.

A tidal residual currents [52] divergence zone close to the release outfall divides waters flowing into the Normandy–Brittany Gulf from the waters forming part of the general flow from west to east up the Channel and towards the Straits of Dover. As a result, small differences in the release conditions can lead to opposite directions of spreading in the medium-term [25,60]. As a consequence, simulation of the dispersion in the area is challenging for the numerical models' dispersion capability.

3.1.2. 3D Dispersion: Super Tidal Time Scale

The La Hague outfall is located on the sea floor, two kilometers off the coast (Figure 5). The vertical dispersion of the plume has been investigated during the DISVER project. In total, 19,000 in-depth samples were taken during three campaigns of four–five days [55]. Figure 5 shows an example of the sampling strategy with plume transects at different distances from the outfall. The transects are slices across the axis of the plume propagation from the surface down to 25 m depth; they are spaced at time intervals of 5 to 10 min. The database encompasses 137 vertical transects up to 65 m depth with 100–200 individual measurements per transect. This provides a rather good picture of the vertical structure of the plume along each of these transects. Figure 6 shows an example of model/measurement comparison during two series of transects at 850–1300 m and 3200–4200 m downstream the outfall. These figures show an accurate representation of the plume in space and time. Figure 6a exhibits unexpected highly complex and variable structures resulting from intense turbulent mixing with eddies of about 100 m that extend to the width of the plume rapidly. The 3D model is not able to reproduce the turbulent mixing at this scale. Further than 3000 m downstream (Figure 6b), the plume reaches the sea surface with a more homogeneous shape that is better reproduced by the model.

The results obtained from all transects have been integrated by the normalization of the concentration measured with the corresponding release. Results are presented in Figure 7 with the measured and simulated extension and dilution of the plume with the distance during a constant release (steady state situation). It shows that if the model does not reproduce the instantaneous turbulence, on average, the shape and dilution coefficient are given the correct order of magnitude. Details of the dilution variations and model/measurement comparisons are given in [55].

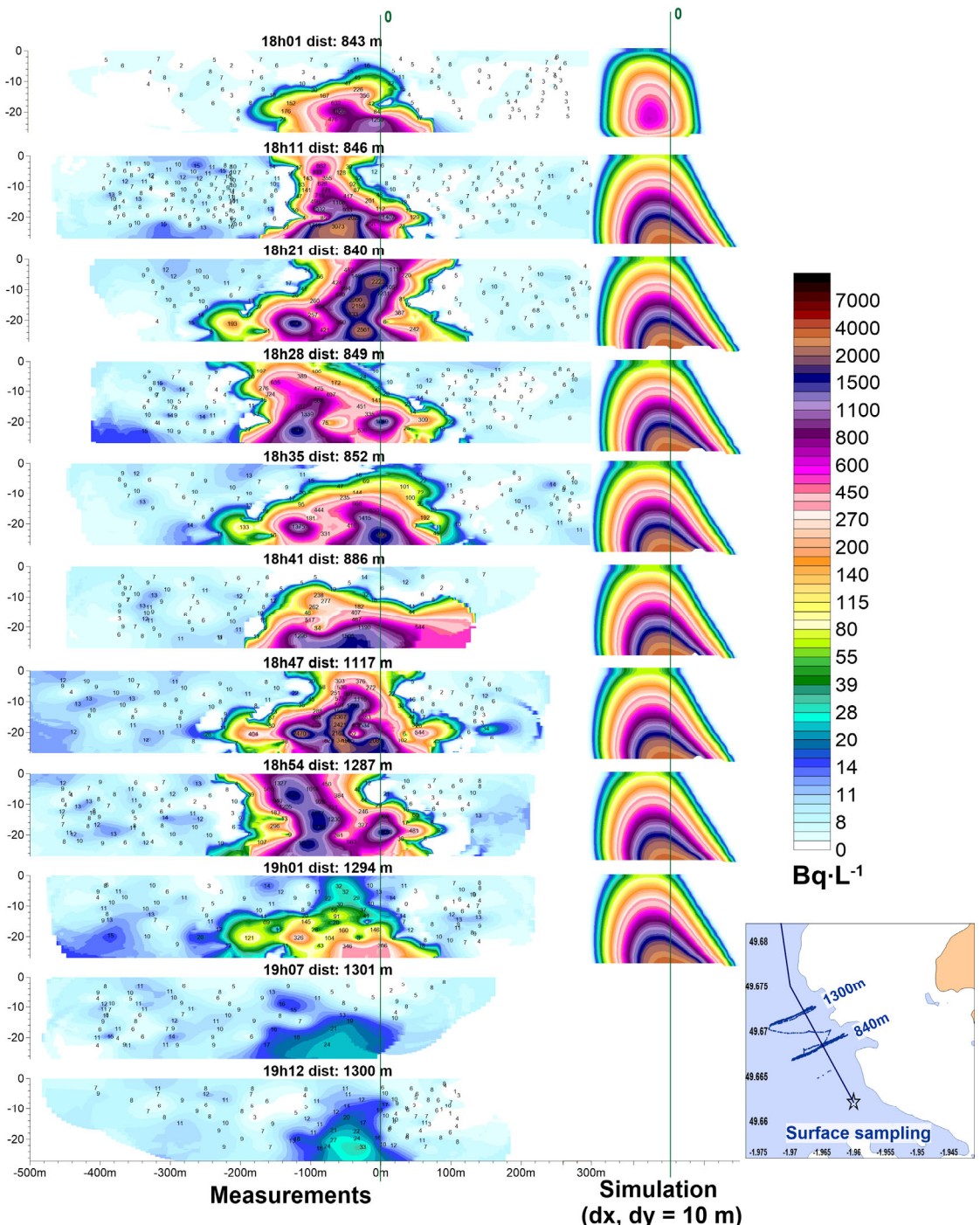

(**a**)

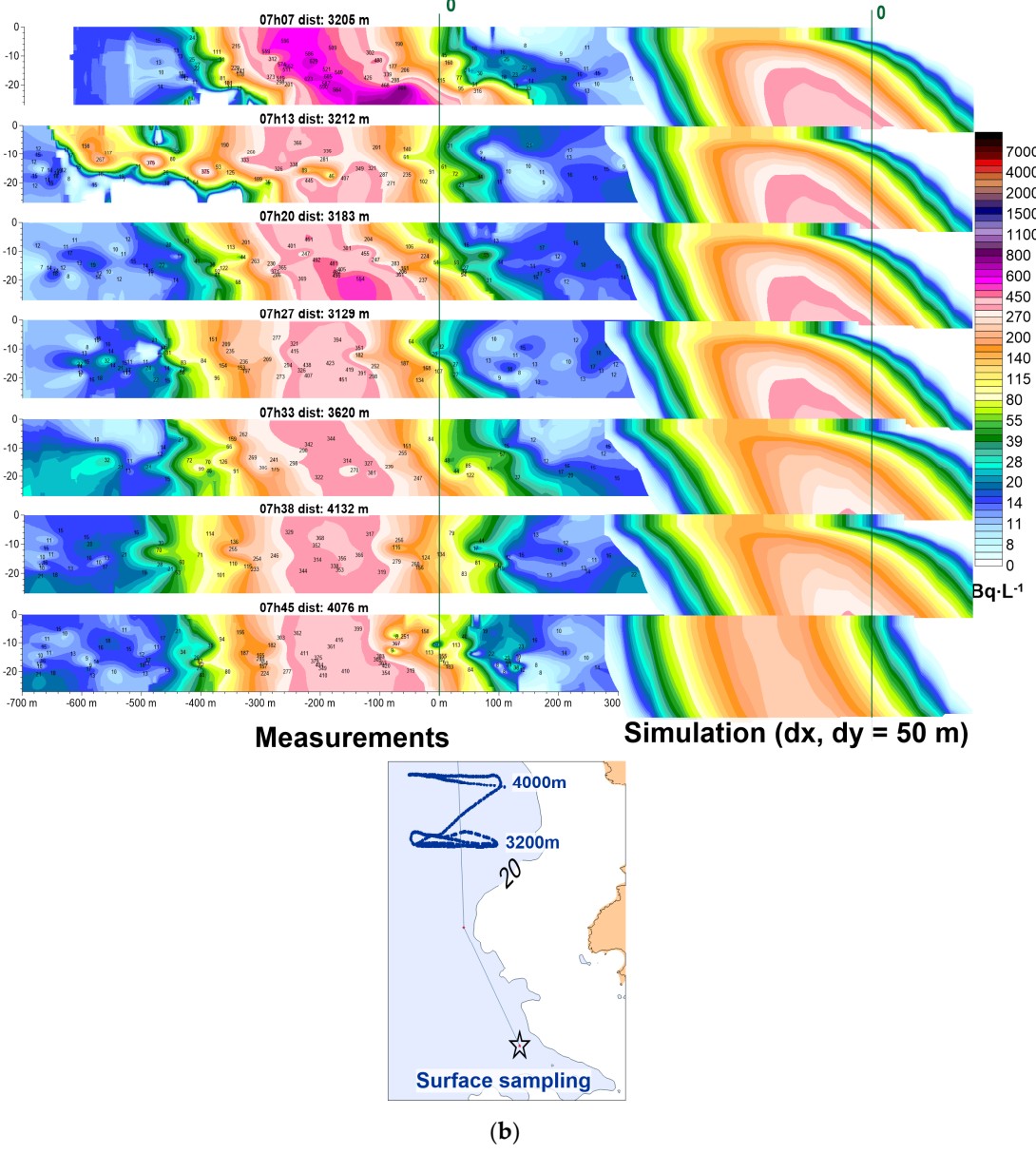

**Figure 6.** Examples of measurement/model comparison of vertical transects of the plume between 850–1300 m distance from the outfall (**a**, 7/10/2010), and 3200–4200 m from the outfall (**b**, 6/4/2011), adapted from [61]. Time and spatial scales (m) are the same for measurements and simulations. Insets: maps of surface sampling locations, adapted from [55].

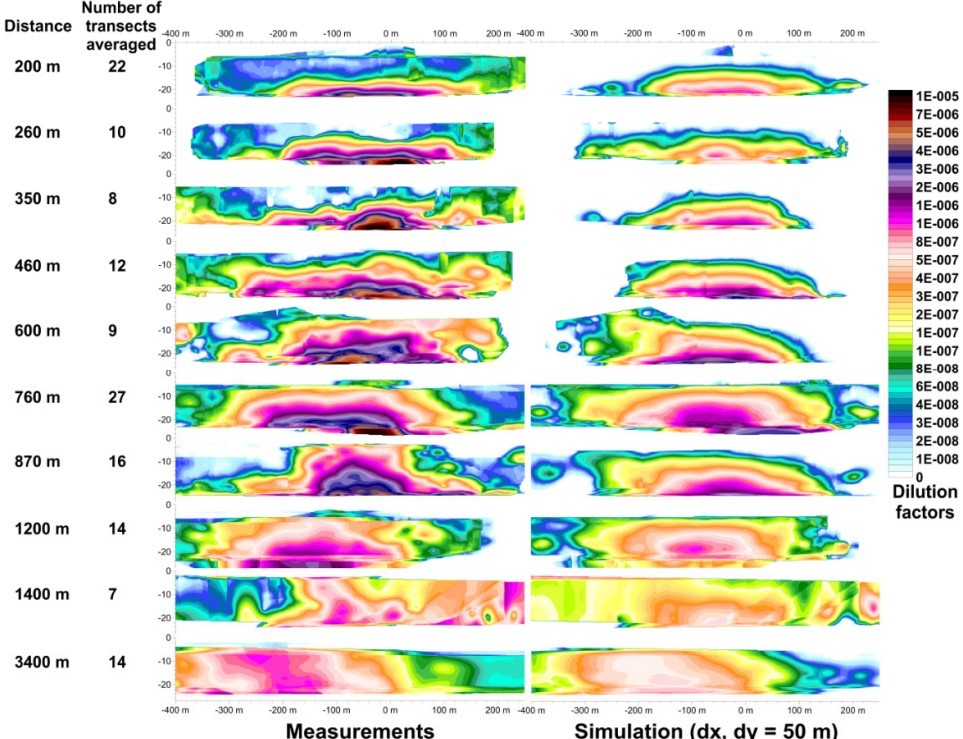

**Figure 7.** Variations with distance of the outfall of measured and simulated average dilution factors, adapted from [55].

As the plume shape is more complex to the north of the La Hague Cape (Figure 5b), a similar model/measurement comparison is difficult to perform. Figure 8 shows the vertical distribution of concentrations 7 km downstream and five–six hours after a release. In this area, the tritium plume crossed the La Hague trough, of which the depths range from 80 to 100 m. These observations result from current shear between surface and bottom waters: the labeling is first transported in surface waters where the currents are larger.

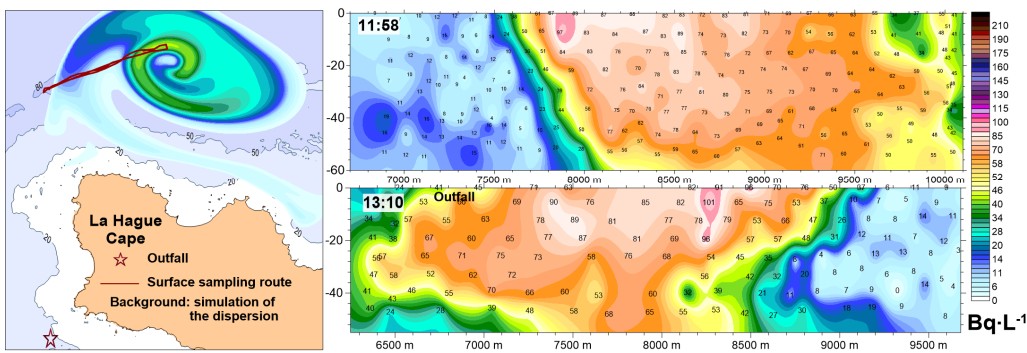

**Figure 8.** Vertical concentrations of the plume five and six hours after a release and at 7 km distance. Left: distribution of surface sampling locations during the survey the 6/4/2011 (red dots). Background: simulation of the expected plume at 12:15. Right: interpolation of in-depth measurements obtained during the survey at 11:58 and 13:10 the 6/4/11 (back and forth). Numbers: concentrations measured at each in-depth location.

### 3.1.3. 2D Dispersion: Tidal Time Scale

As shown in Figure 7, the vertical homogenization of the plume is reached at 3 km downstream the outfall, which corresponds to one hour after the release. Except at particular locations where

strong bathymetry changes occurs (Figure 8), in the eastern English Channel, surface measurements of radionuclide concentrations are representative of the whole water column labeling. Investigations have been performed to test the model capability to simulate the dispersion of the La Hague plant within 1 to 48 h after the beginning of the release (DISPRO project). Beyond 48 h, it is no longer possible to distinguish two daily consecutive releases from another, because dilution and stirring mix them together. Within 48 h following a release, about 200 transects were done with sampling every 30 seconds (i.e., transects crossing a dispersion plume that could be traced back to a known release, and into which the width of the plume could be assessed). These transects represent roughly 3000 individual measurements, which encompass all tidal conditions (from neap to spring tides). An example of the tracking of one release together with a model/measurement comparison is given in Figure 9. Comparisons have been performed by accounting for the measured and simulated concentrations at a sampling location, and other criteria described in Section 2.5.1. Additional comparison criteria. Details of the results obtained are in [26]; Table 2 presents the main results of model/measurement comparisons for short scales model/measurements comparisons. For example, at short scales next to an outfall, the deviation between measured and simulated concentrations (expressed as mean dilution coefficient) is lower than 71% for a 95% confidence interval (Table 2).

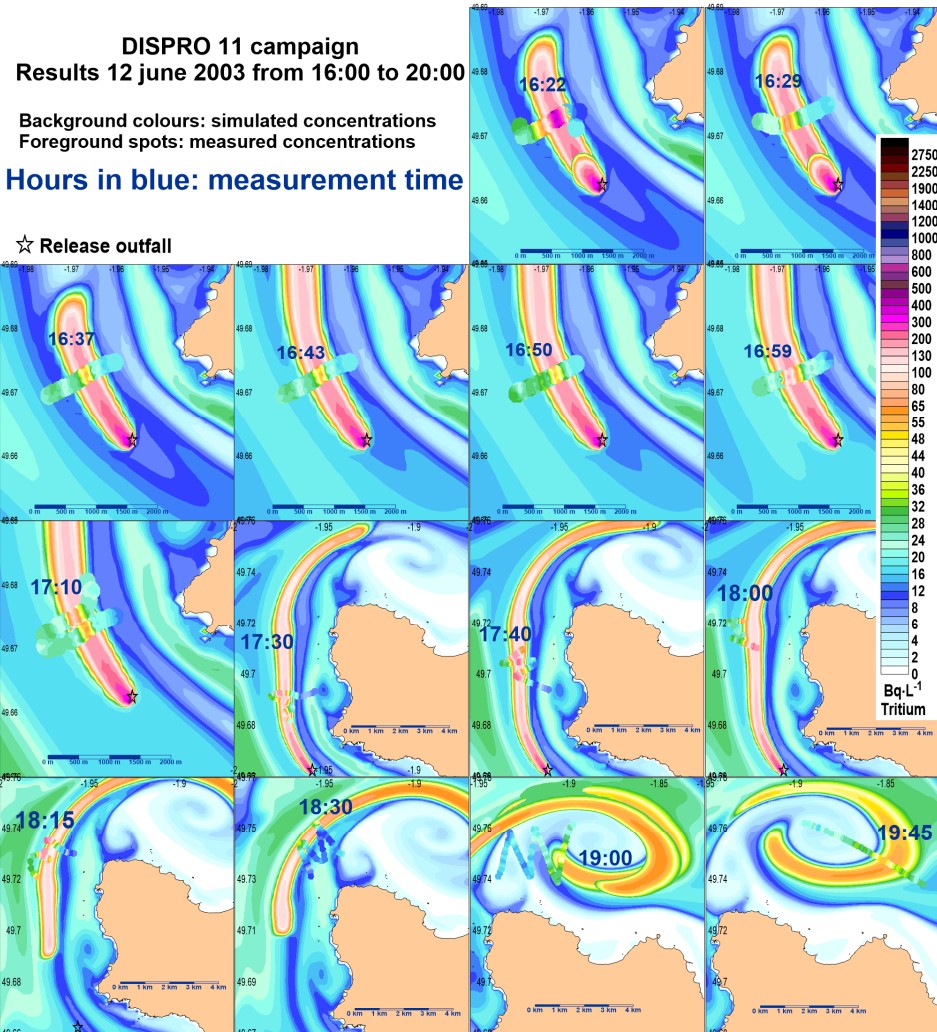

**Figure 9.** Example of a comparison between the measured and simulated dispersion during 4 h after release, from16:22 to 19:45 adapted from [25].

**Table 2.** Characteristics of the model used to simulate soluble radioactive release dispersion in seawaters at a short scale next to an outfall around the Hague Cape.

| Dispersion Characteristics 1h to 48 h after Release | |
|---|---|
| Geographical boundaries of the model | 49°17′ N–49°55′ N; 2°26′ W–1°31′ W |
| Hydrodynamic model | Two-dimensional: Vertically-averaged velocities and concentrations, 110 m mesh size, 20 s time step |
| Average discrepancy between the mean dilution coefficients measured and simulated in the plumes | 9% (−66% → 70%) |
| Average discrepancy per transect between simulated and measured maximum dilution coefficients | 3% (−72% → 73%) |
| Average measured/simulated plume-width discrepancy | −6% (−73% → 65%) |
| Average discrepancy between measured and simulated plume-position, as a function of distance from the outlet point | −1% (−22% → 22%) |

Figures in brackets indicate the 95% confidence interval.

*3.2. Large Scale Model/Measurement Comparisons: Multi Tidal Time Scales*

3.2.1. Individual Measurements

When a sufficient amount of measurements was obtained during oceanic campaigns, we could draw up radionuclide concentration maps at the scale of each survey. Campaigns performed repeatedly covered the English Channel and the North Sea between 1988 and 1996, the north-west Atlantic and Irish Sea between 1994 and 1996, the North Atlantic between 1997 and 2004 and the Bay of Biscay between 2009 and 2016. Hydrodynamic models accounting for all source terms and oceanic forcing (tide, meteorological forcing and large-scale circulation and hydrology) allow comparing of the measured and simulated concentrations at the date of each campaign. Figure 10 gives an example of such a comparison in 1994 for [125]Sb and 2016 for [3]H.

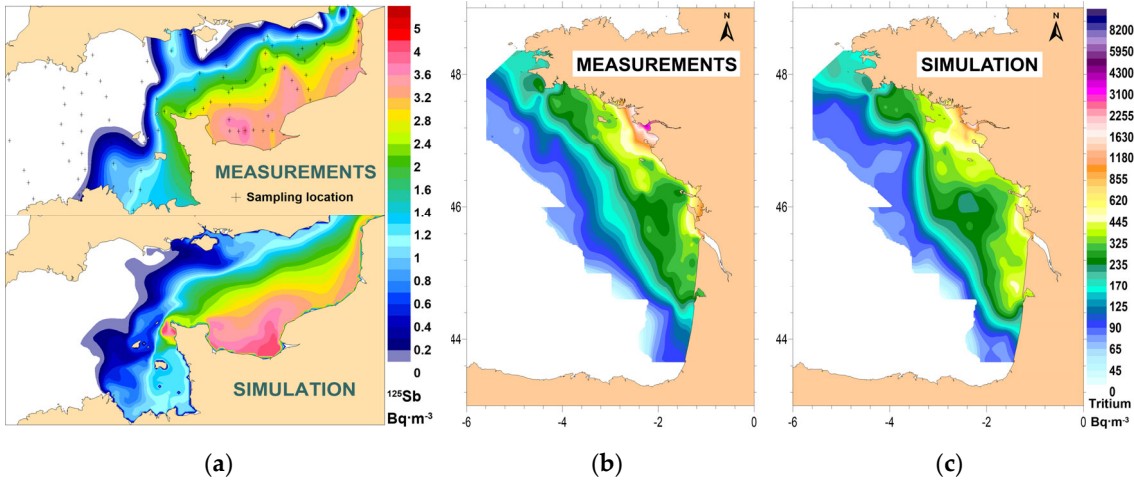

**Figure 10.** (**a**) Measured and simulated [125]Sb in the English Channel, September 1994, adapted from [24]; measured (**b**) and simulated (**c**) surface [3]H in the Bay of Biscay, spring 2016, adapted from [26].

Computation of the measured and simulated concentrations at each sampling location and time gives statistics of the model representativeness. For example, in the English Channel and the North Sea, application of a residual Lagrangian model gives a mean difference between the 1400 individual values calculated and measured between 1988 and 1994 of 54% [24]. The deviation at a short scale next to an outfall is lower than 70% (Table 2). At the scale of the Biscay Bay with a 3D model the deviation is 21% [62].

### 3.2.2. Water Masses Labeling

In case of multi-tracer studies, it is possible to associate a specific labeling of the different water masses investigated and to draw up a "picture" of the different plumes associated to the different water masses. This means that a set of initial average concentrations is affected at each water mass entering the studied area. Then, at each sampling location in this area, the resolution of an equations system can determine the contribution of each water mass that fits the measured concentrations of tracers. This implies that the number of unknown (water mass to account) does not exceed the number of independent tracers measured.

This method was particularly fruitful in 1988 in the North Sea, where it was possible to distinguish and map the distribution of the four main water masses entering the North Sea by using the simultaneous measurements of three radionuclides ([125]Sb, [137]Cs, [134]Cs) and salinity (Figure 11) [17]. As compared with another campaign performed two years before, the contribution of the direct fallout after the Chernobyl accident and the rate of renewing of North Sea waters were also calculated.

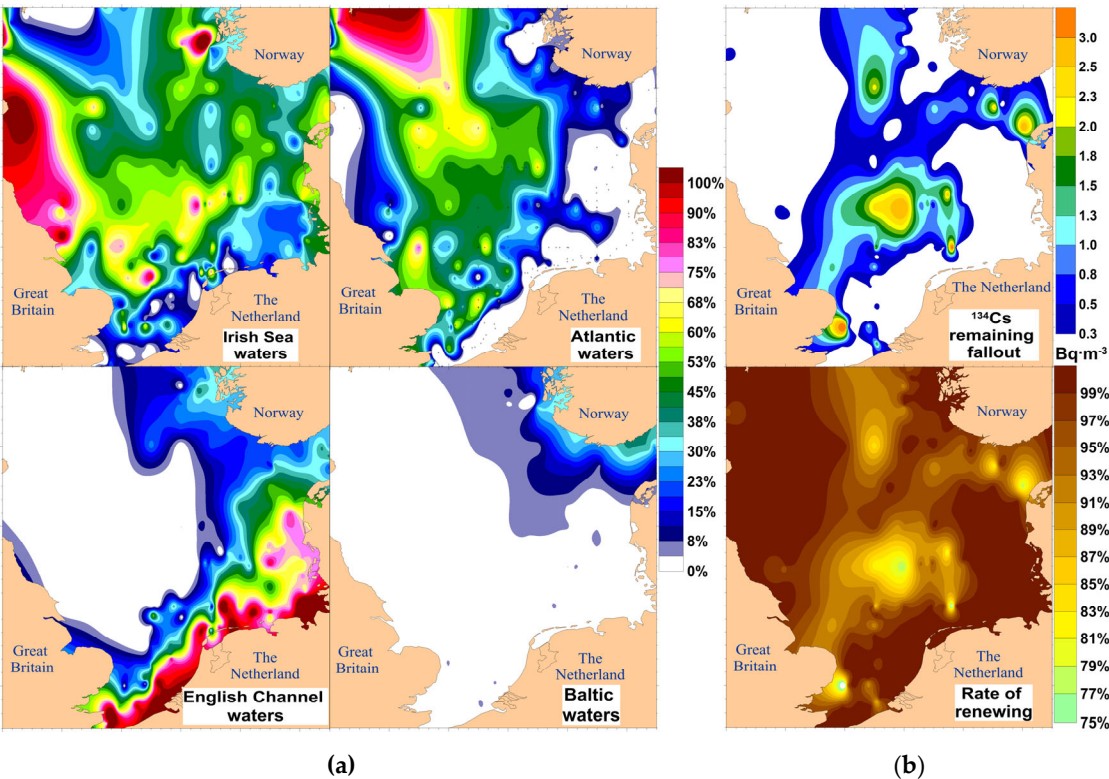

**Figure 11.** Percentage origins of water masses entering the North Sea in 1988 deduced from dissolved radionuclides measurements (**a**); [134]Cs remaining fallout from the Chernobyl accident and rate of renewing of water masses in two years (**b**), adapted from [17].

A similar approach was applied in the Bay of Biscay in 2016, by accounting for the different [3]H labeling of waters coming from the Loire and Gironde rivers. Therefore, it is possible to compare

directly the extent of the plumes deduced from measurements and simulated separately as shown in Figure 12 [40].

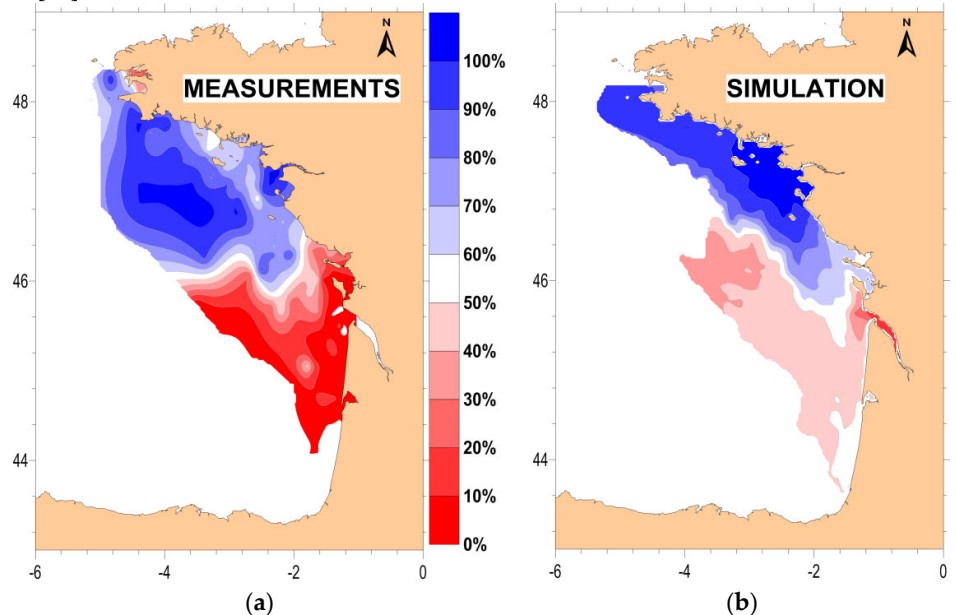

**Figure 12.** (**a**) Measured and (**b**) simulated freshwater contribution of the Loire River relatively to the Gironde River, adapted from [26] in spring 2016.

### 3.2.3. Integration of Normalized Contributions

In a similar way as presented at the end of Section 2.5.1., it is possible to compare the plume distributions obtained from different campaigns by normalizing the measured concentrations with the corresponding fluxes of releases. This method allows the average dispersion characteristics and dilution coefficient of the considered source term to be mapped. It has been applied on a short scale in 3D (Figure 7), in the English Channel [20] and the North Sea [24], as presented in Figures 13 and 14. It shows that, on average, the Lagrangian residual models are able to properly catch the dispersion process at that large scale. This result was obtained after adapting the wind stress drag, which is one of the main hydrodynamic model calibration parameters.

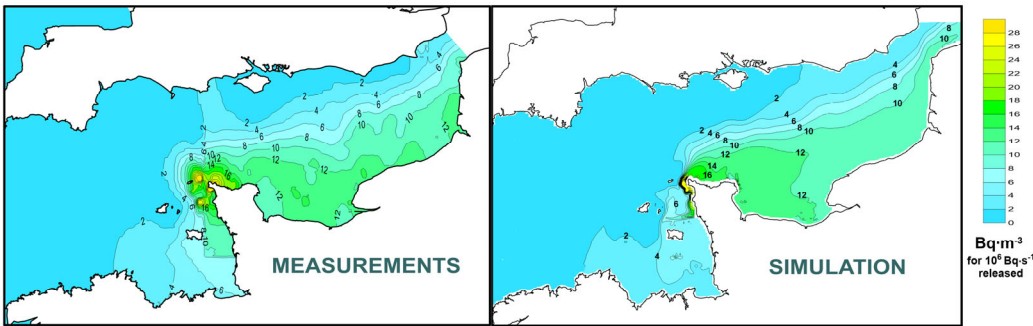

**Figure 13.** Normalized measured and Lagrangian residual simulated distribution of $^{125}$Sb between 1983 and 1994 for a constant release of $10^6$ Bq·s$^{-1}$, adapted from [20].

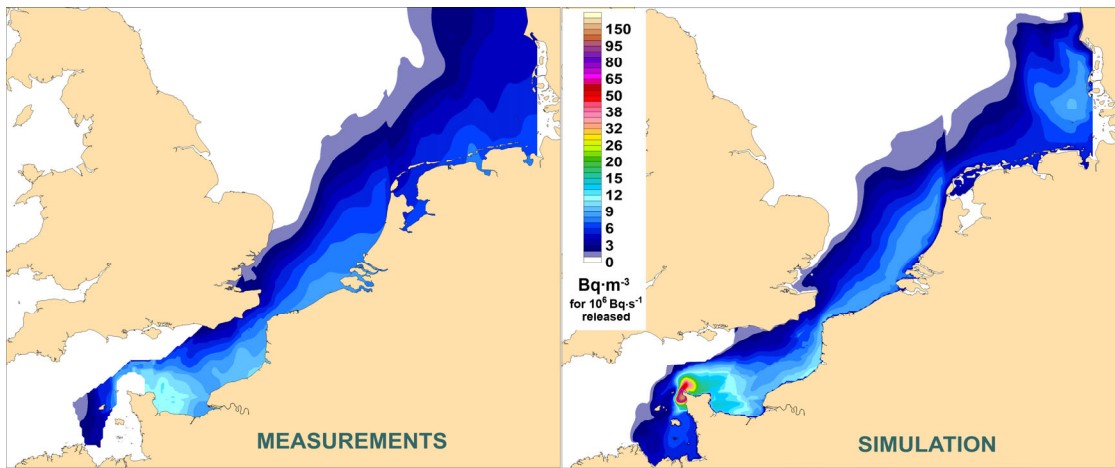

**Figure 14.** Normalized measured and simulated distribution of $^{125}$Sb between 1988 and 1994 for a constant release of $10^6$ Bq·s$^{-1}$, adapted from [24].

### 3.2.4. Inventories

As presented in Section 2.5.2., inventories of radionuclides quantities in different areas and water masses allow for the assimilation of numerous measurements in integrated quantities to account for the labeled depth and distribution of samples. The results could be compared with the known releases supposed to contribute to the labeling, and to model results in the same areas and water masses. Comparison with releases without simulation have been performed in the North Sea by slicing the southern North Sea into boxes where measured radionuclide quantities were compared with the La Hague releases [18] (Figure 15). From this correspondence it was possible to highlight the average transit time (one year from La Hague to the Skagerrak) and fluxes through the Dover Strait (97,000–195,000 m$^3$·s$^{-1}$), as well as estimations of losses for non-conservative radionuclides [18].

In the English Channel and the southern North Sea, the equilibrium between the measured and simulated quantities of radionuclides was an efficient tool to tune the wind friction that provide the best reliability of the Lagrangian residual model [24].

The loss of non-conservative radionuclides from seawater to sediments and living species (around 15% for $^{134}$Cs, 75% for $^{106}$Ru and 85% for $^{60}$Co Table 3, Figure 16) was assessed from the measured quantity comparisons with the corresponding releases at the scale of the English Channel. It is then possible to check the environmental models accounting for the fluxes between the different environmental compartments. Such balanced budget exhibited an unknown source term at this scale, which represented twice the expected quantities measured for $^{137}$Cs at the Channel scale (Table 3).

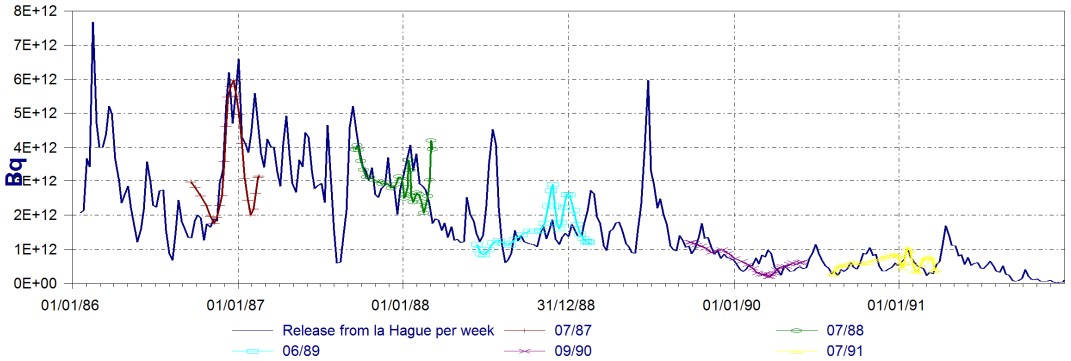

**Figure 15.** Quantities of [125]Sb measured in North Sea boxes during oceanographic campaigns compared to the corresponding releases from the La Hague plant, adapted from [18].

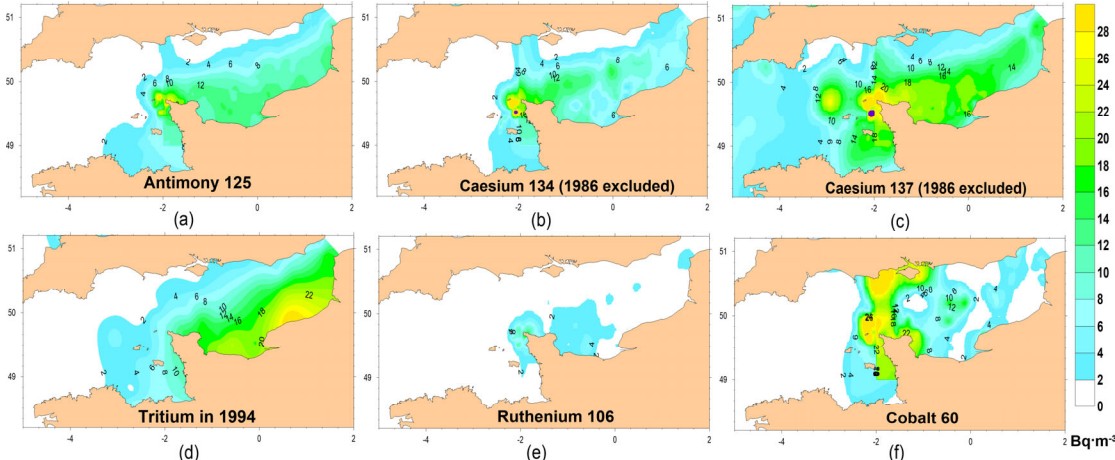

**Figure 16.** Average impact of the La Hague reprocessing plant in seawater corresponding to a constant release of 1 MBq·s[−1]; **(a)** [125]Sb, **(b)** [134]Cs, **(c)** [137]Cs, **(d)** [3]H, **(e)** [106]Ru, **(f)** [60]Co, adapted from [20].

**Table 3.** Comparison of measured quantities of radioactivity in the English Channel with correspondent releases from La Hague from 1983 to 1994, adapted from [20]. Antimony-125 is used as reference for the calculation; the background concentrations in Atlantic surface waters ([137]Cs, [3]H) and Chernobyl fallout ([137]Cs, [134]Cs) has been deduced.

| Volume: Equivalent Release Duration: | Whole English Channel 4702 Km³ 32 Weeks–7.3 Months | | | | | | Eastern English Channel 1576 Km³ 25 Weeks–5.7 Months | | | | | |
|---|---|---|---|---|---|---|---|---|---|---|---|---|
| | [137]Cs | [134]Cs | [106]Ru | [125]Sb | [60]Co | [3]H | [137]Cs | [134]Cs | [106]Ru | [125]Sb | [60]Co | [3]H |
| Number of campaigns | 3 | 2 | 3 | 3 | 3 | 1 | 5 | 3 | 5 | 5 | 2 | 1 |
| **Fraction of the La Hague release** | **233%** | **86%** | **26%** | **98 %** | **14%** | **103%** | **139%** | **83%** | **19%** | **98%** | **8%** | **121%** |

We hypothesized [18] that the [137]Cs excess in the English Channel resulted mainly from the influence of Sellafield releases. It was afterwards confirmed [11]; this evidences a pathway from the Irish Sea through the St George Channel in the Celtic Sea, and then a seasonal input in the English Channel as shown in Figure 1. The flux that reaches the English Channel represents around 1% of Sellafield [137]Cs. Figure 17 shows the English Channel areas that are the most impacted by the Sellafield releases. This corresponds to locations where sediments act as a delayed secondary source term for the water column, as demonstrated in the Irish Sea [63]. It concerns mostly the Hurd deep in the English Channel center and the French coastal areas.

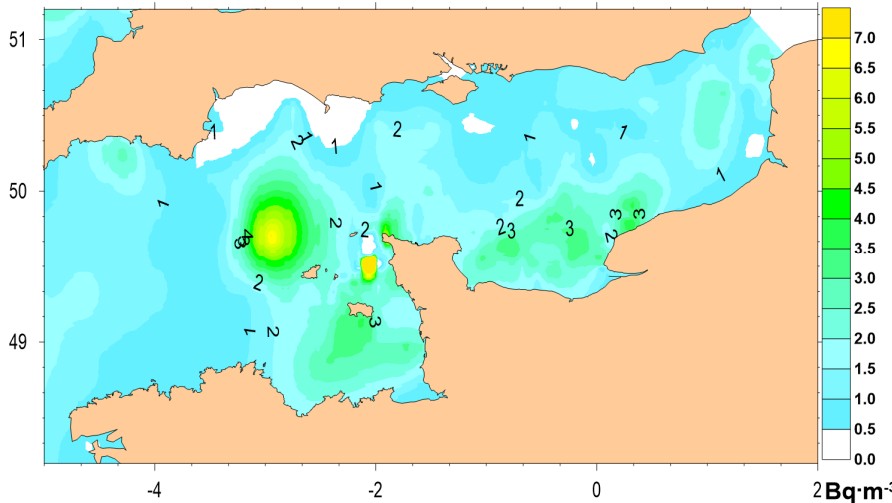

**Figure 17.** Calculated from Figure 16 ($^{137}$Cs–$^{134}$Cs): $^{137}$Cs not coming from La Hague (average during 1986–1994).

In the Celtic Sea, comparisons between inventories and releases from La Hague and Sellafield plants give an estimate of the fluxes and residence times of water masses in the Celtic Sea and North Est Atlantic approaches, with labeling from Sellafield releases along the western Ireland coasts [11] Figure 1.

At a global scale, a comparison of the measured $^3$H inventories and known inputs have been applied at the world oceans scale (Figure 18 [64]). It provides fluxes, estimations of residence time and values for the past and future concentration of $^3$H in Atlantic waters entering the European waters (78 Bq·m$^{-3}$ in 2016).

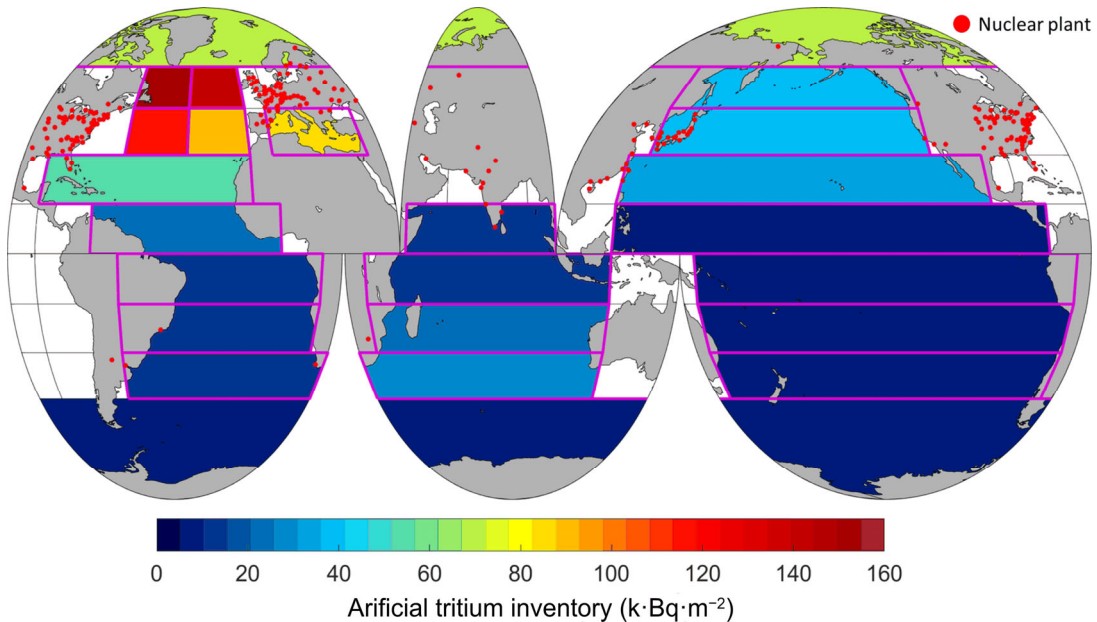

**Figure 18.** Inventory of anthropogenic tritium by unit of surface area (kBq·m$^{-2}$), adapted from [65].

This value was applied to compare the measured and simulated radionuclide inventories in the Bay of Biscay. At this scale, the concentration simulated by the MARS3D model was compared to the measured ones by accounting for the variation of the $^3$H concentrations with depth [26]. It provides an estimation of the residence time (around one year for the continental shelf of the Bay of Biscay), and pathways of waters coming, respectively, from the Loire and Gironde rivers [26].

Estimation of radionuclides inventories was also applied during the accidental context following the nuclear accident at Fukushima Dai-ichi power plant. Many seawater measurements of [137]Cs were provided continuously by Japanese authorities (Figure 19).

They enable the calculation of successive inventories of [137]Cs quantities in the surrounding Pacific waters and to estimate the marine source term coming at sea (Figure 20) [58]). This first estimation of 27 PBq (12 PBq–41 PBq) appeared to be very high at the time of publication, but later estimations were in the same order of magnitude [65–67]. The rapid decrease of [137]Cs quantities during the weeks following the accident reveals a very rapid environmental half-life that has been reproduced by MARS3D simulation (Figure 20 [27]).

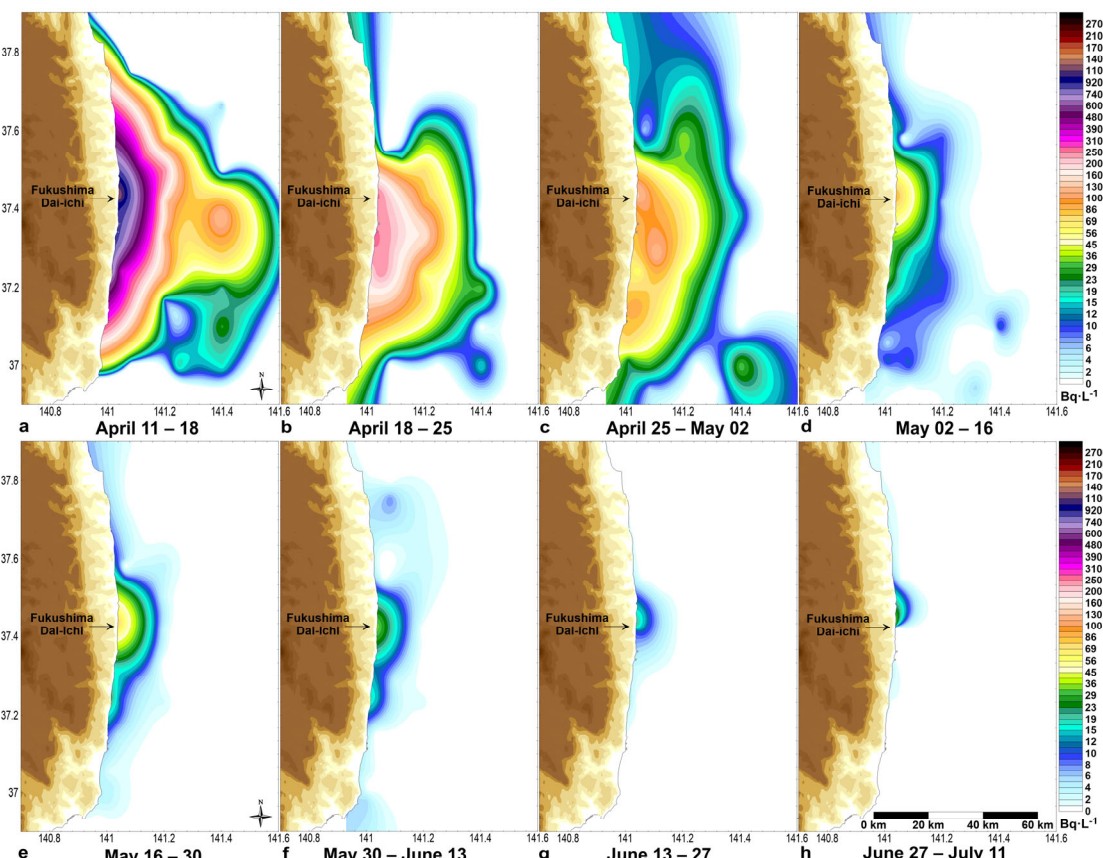

**Figure 19.** Concentrations of [137]Cs in seawater between 11 April and 11 July 2011, adapted from [59].

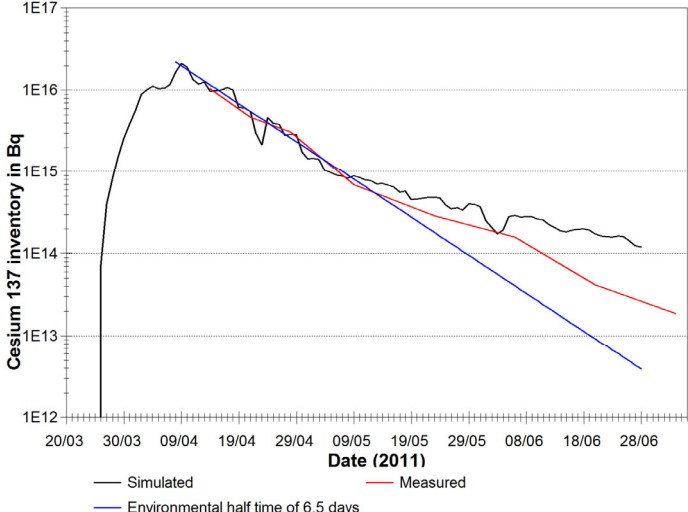

**Figure 20.** Time evolution of measured and simulated [137]Cs inventories present in the Fukushima area, adapted from [27].

## 4. Discussion

This work gives accessibility to a comprehensive database, including in-situ radionuclide measurements performed during oceanographic campaigns and fluxes of dissolved radioactive release between 1982 and 2018 over the North-Western European continental shelf.

The results presented show applications of in-situ radionuclide measurements coupled with hydrodynamic models. They demonstrate how they improve the knowledge of dispersion in seawater. Applied alone, in situ data provide general pathways, transit time and dilution rate at different scales. Coupled with numerical hydrodynamic models, they improve and confirm their reliability with strong constraints. Physical measurements (as current, sea level or drifter measurements) generally represent short term parameters. Dissolved radionuclides complement physical data with information associated from short to long term transport and dispersion in seawater that could not be directly measured in currents.

Use of radiotracers for model validation was initiated by Salomon and Guéguéniat in 1988 with residual Lagrangian models [68]. This pioneering work was applied at different scales with space and time adaptations to obtain results comparable to measurements and simulations.

Artificial dissolved radionuclide tracers are particularly interesting because they fulfill a detection limit close to the background level, and contain few well known source terms and low risk of pollution during sampling and treatment. These advantages allow the detection of labeled plumes from short (hours, 100 m) to large scale (years, 1000 km). It counterbalances the efforts given to radionuclide measurements such as sampling large volumes for gamma emitters and measurement constraints.

### 4.1. Radiotracers

The use of artificial radionuclides such as oceanographic tracers depends from the controlled liquid releases performed by nuclear industry. These releases represent opportunities of existing labeling that could be used to follow water masses. They are issued mainly from European reprocessing plants. The fluxes of gamma emitters have decreased by two orders of magnitudes during the 1990s after application of a more efficient industrial purification process to diminish the environmental risk. It results in that the concentrations measured in the vicinity of the La Hague plant in 2018 are close to the detection limit and lower than the ones measured at 1500 km distance along the Norwegian coasts in 1988 (less than 2 Bq·m$^{-3}$). Other radionuclides could still be applied as long term tracers, such as tritium [11,26,64], [99]Tc [69] [129]I [70,71] or [236]U [72–74].

[3]H is present in all nuclear plant releases and, in the form of tritiated water HTO, it could not be retained at the source by chemical process. At the nearest monitoring station to the main outfall in the marine system at La Hague Cape, concentrations in seawater are on average 12 Bq·m$^{-3}$ (2017–2019, O. Connan pers. comm.). This concentration is three orders of magnitude lower from the OMS maximum recommended value of 10,000 Bq·L$^{-1}$ for drinking water. The [3]H labeling from the nuclear plant will remain significant as long as the nuclear plants operate, and thus, they remain a relevant long term oceanographic tracer if we have the capability to measure low levels (from the oceanic background of 0.07 Bq·L$^{-1}$ [64]). The mapping of the dispersion of the Loire and Gironde waters at the scale of the Bay of Biscay demonstrates this approach [26]. Limitations for radiotracers measurement concern mainly the available means (low level counters, [3]He degassing and mass spectrometry) and delay between sampling and measurement (from days to years). As an example, the [3]He ingrowth method for [3]H measurement exists in few laboratories around the world.

### 4.2. Hydrodynamic Models

Oceanographic hydrodynamic models have been continuously improved following the computing capabilities enhancement. It is possible to represent the whole continental shelf in 3D with a resolution of around one kilometer. This calculation efficiency must not mask the requested improvements concerning the knowledge and representation of physical process. The representativeness and precision of forcing the parameters determines the model reliability.

It concerns open boundaries limit conditions (stratification, currents, tides), and local data such as bathymetry precision, meteorological effects, bottom nature and associated friction. A better understanding of the surface and bottom drag coefficients remains an ongoing concern in research [75], as they require significant improvements.

A demonstration that a given hydrodynamic model is reliable does not prove that other similar models are at the same level. Long term advection and dispersion are particularly discriminant, as it represents only a few percent of the instantaneous currents. The radiotracer databases will remain useful to test the next hydrodynamic model generation.

Nevertheless, the level of realism reached between measurements and simulations give way to hydrodynamic model application at the scale of the European shelf or elsewhere in the world as shown in the north-western Pacific [27].

For short scale studies in 3D, MARS3D is not able to represent the real turbulence process (Figure 6). These phenomena are outside the initial scope of this model (i.e., regional scale); it must be improved by applying other numerical schemes or discard assumptions as the hydrostatic one. The computing capabilities remain a limit to simulate together short scales (meters) and longer ones (tens of kilometers); however, methods exist to overcome these limits (AGRIF) [76].

### 4.3. Scales for Model/Measurement Comparisons

This work shows that different approaches must be applied to measure the dispersion from short to medium or large scales. An operational definition to distinguish short scale from longer ones in tidal seas is the capability, or lack thereof, to distinguish each individual release plume. This depends from the diffusion process and the frequency of the main forcing parameter and releases. This results in that in the center of the English Channel, short scale studies concern fewer than four tidal cycles (two days).

The short scale requires models simulating rapid plume displacements (30 km in 3 h) with a high resolution and short time step (lower than 100 m and 20 s). A difference of 10 minutes or 200 m could change the punctual concentration close to the plume of one order of magnitude. If we consider that the model simulates the tidal propagation over thousands of kilometers before calculating the local tide, this implies strong constraints on the numerical scheme and forcing parameters (open boundaries conditions and drag coefficients). The dispersion of individual releases could be measured, but the rapid changes during dispersion do not allow drawing 3D or 2D maps that describe the plume dispersion several times per hour. 2D and one-dimensional (1D) comparisons were performed for 3D and 2D models comparisons, respectively.

The more the model/measurement comparison is done far from a source term, the more the plumes are smoothed and the lower the deviations between the punctual measurement and equivalent simulated values. Larger scales could use, when tidal effects are smoothed, instantaneous or Lagrangian residual models with a mesh size larger than one kilometer and time steps higher than 200 s. Exhaustive measurements could be performed that allow calculating inventories with possibility to compare measured, simulated and released radionuclides quantities.

Long term-large scale models obtain good model/measurement scores without under- or overestimation of radionuclides concentrations if they account well for the residual effects of the different influences (tidal, meteorological, frictions). These influences can be adjusted such as the wind drag coefficient obtained in [24]. The determination of detailed physical phenomena hidden behind these calibration parameters is an open research field.

### 4.4. Sampling Tools

The measurement of radionuclide dispersion led to the development of original systems to sample seawater in highly dynamic environments. High frequency samplers allow the simultaneous collection of 10 samples each 30 s while recording the time, location, volume, flow and depth [39]. The different parts of the developed tools may be used separately or together and adapted for other purposes.

Volumes collected could be extended or sampling depth increased that allow continuous in-depth sampling of any dissolved substances with a vessel sailing. 3D sampling is not limited to seawater and could concern larvae or particulate matter with continuous sampling and measurement of vertical suspended particulate matter (SPM) profiles.

This enhances the efficiency of oceanographic campaigns. As an example, the towed depressor Dynalest was applied as a support for instruments during sailing: an acoustic current doppler profiler (ADCP) was deployed to measure currents from 3 m depth up to the bottom without being perturbed by surface waves with a ship sailing normally in the Alderney race (0.5–4 m.s$^{-1}$).

Variability of plume dispersion made short scales studies impossible without accurate simulation tools to forecast its location. Sampling assisted by model simulation has demonstrated its efficiency in the vicinity of an outfall in a tidal environment (Figure 5 and Figure 9, [25,39]).

### 4.5. Radionuclides Inventories

By accounting for all available measurements associated with their representative area and depth, it is possible to map the radionuclide distributions and calculate radionuclides quantities existing in specific areas. Such inventories give integrated values with a better precision than individual concentrations [17,20].

Marine dissolved radionuclides give a rare example in environmental studies where balance budgets could be calculated between measured, simulated and releases inventories with precision better than 10% [25,26,20]. Comparison between radionuclides inventories quantities existing in seawater with the known sources results in determinations of environmental fluxes and transit times [18,26]. It highlights unknown sources and water masses pathways [11,20,58]. Inventories could also be compared with the simulation results and they represent robust check values to improve hydrodynamic models [20,24,27].

The balance obtained between released and simulated quantities of conservative radionuclides gives access to the loss of less conservative radionuclides. It is possible to quantify the fluxes between the different environmental compartments (i.e., seawater, living species and sediments) [18,20]. Radionuclide inventories are then powerful tools to check integrated environmental models accounting for stocks and fluxes of radionuclides at the scale of a continental sea as the English Channel.

### 4.6. Normalization

Normalization by a parameter representative of an environmental or anthropogenic forcing could be applied to long term time series or variable spatial distributions. This method made possible, by accounting for adequate time or space scale, to compare highly variable temporal or spatial changes of the environment as a result of variations depending on the season, meteorological effects, tides and releases. Normalization mitigates stochastics effects and highlight average characteristics of the environment difficult to quantify. It gives useful results with the measurements alone, but can also be compared with simulations. Others examples are: 3D short scale dispersion characteristics (Figure 7); average distributions and inventories in the English Channel (Figure 16 and Table 3) and North Sea (Figures 14 and 15); shape of the English Channel pathway in the southern North Sea [18]. Similar methods could be applied in areas where deterministic models fail to represent the water masses circulation as in the Kuroshio gyres in the Northern Pacific [77].

### 4.7. Perspectives

Tracers and methods applied here are potentially applicable elsewhere if the radionuclide labeling is sufficient to distinguish different water masses. This is obvious in the case in the Irish Sea, where the other main artificial source is the Sellafield reprocessing plant. The work performed in the Bay of Biscay demonstrates that nuclear power plant releases of tritium are sufficient to label water masses at a large scale if low level measurement methods are available [26]. This was also possible after precise evaluation of the North Atlantic surface water background concentration [64].

The possible application of $^3$H as water masses tracer potentially concern all seas reached by nuclear power plant releases. Maps of power plant locations with the corresponding tritium releases (Figure 18 [64,78]) suggest the Mediterranean Sea, North-West Atlantic, Arctic Sea, Western and Eastern Pacific, Black Sea, Baltic Sea, China Sea, Japan Sea, Gulf of Mexico and Arabian Sea.

Radionuclide dispersion models were applied to simulate realistic behavior of controlled industrial releases from 1984 up to 2016 in the English Channel (Figure 21). Such results could be compared with monitoring measurements and give a way to understand and interpret them. It was applied by IRSN to optimize the location of monitoring stations. Another application is the forecasting of accidental situations as it was performed after shipwrecks in the English Channel (Ievoli Sun in 2000, Ece in 2006) or after the Fukushima accident [27]. An operational tool is in course of development at IRSN to forecast the consequences of accidental situations (STERNE tool [79]).

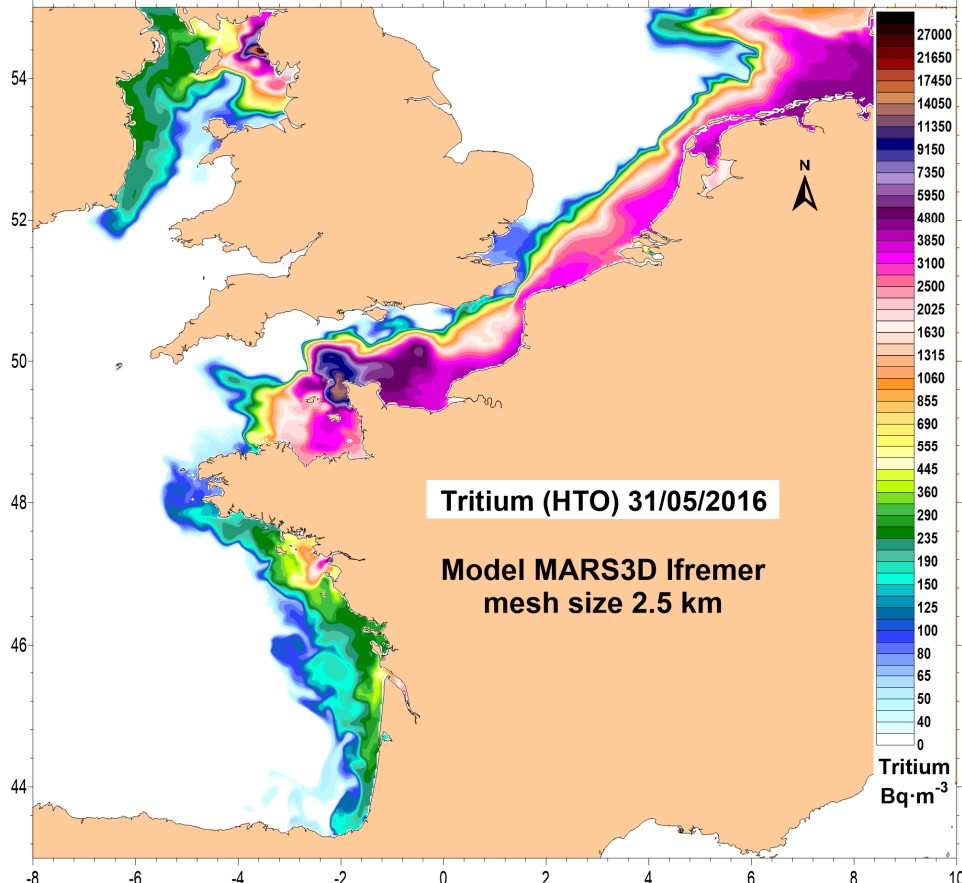

**Figure 21.** MARS3D simulation of the dispersion of tritium industrial releases in seawater over the European continental shelf.

Knowledge of dissolved transport (currents, advection and dispersion) is the basis of other oceanographic studies. It made possible pollutant transfer studies between seawater and living species [56,80–82] or particulate matter and physical transport of sediments [83–85]. Radio-ecological models accounting together for biological and hydro sedimentary process are accessible. The first tests have also shown the possibility to follow the sea/atmosphere exchanges by using $^3$H in water vapor [86].

Radionuclides behavior could be studied at the earth scale [65], as a tool for testing climate changes models (renewing of ocean water masses with $^3$H, water cycle, carbon cycle with $^{14}$C). Radionuclides were widely applied during the 1980s, but are not outdated environmental tracers.

Our wishes are that the provided database will contribute to these works and will be complemented.

**Supplementary Materials:** The following are available online at: Bailly du Bois, P. IRSN measurements of dissolved radioactivity in seawater, 1982–2016, 39642 stations. Available online: https://doi.pangaea.de/10.1594/PANGAEA.906541 (dataset in review, waiting for publication acceptation) (accessed on 02 October 2019). Bailly du Bois, P. IRSN measurements of dissolved radioactivity in seawater, 1982–2016, including liquid releases database issued from British and French nuclear plants. Available online: https://doi.pangaea.de/10.1594/PANGAEA.906749 (dataset in review, waiting for publication acceptation) (accessed on 2 October 2019).

**Author Contributions:** Conceptualization and Methodology, Formal analysis, P.B.d.B.; Data curation and databases, P.B.d.B., C.V. and P.-E.O.; Funding acquisition, P.B.d.B.; Investigation, P.B.d.B., F.D. and P.-E.O.; Project administration, P.B.d.B.; Software, M.M. and P.B.d.B.; Supervision, P.B.d.B.; Validation, P.B.d.B. and P.-E.O. Visualization, P.B.d.B. Writing—original draft preparation, P.B.d.B. Writing—review and editing, P.B.d.B. and F.D. Oceanographic campaigns, P.B.d.B., C.V., M.M., F.D. and P.-E.O. Radionuclide measurements,

L.S., P.B.d.B., C.L. and C.V. Model development, F.D. and P.B.d.B. Technical development, P.B.d.B. All authors have read and agreed to the published version of the manuscript.

**Funding:** Most of the oceanographic campaign uses vessels and crew provided by the French oceanographic fleet operated by CNRS/INSU (Centre National de la Recherche Scientifique, France), Ifremer-Genavir (Institut Français de Recherche pour l'Exploitation de la Mer, France) and GEA-Navy (Groupe d'études atomiques de la Marine nationale). British "Cirolana" and German "Validivia" oceanographic vessels participate also to the sampling collection. This research was partly funded by Orano Company for oceanographic data acquired between 1990 and 2011.

**Acknowledgments:** It is not possible to mention all people that have contributed to the collection of samples during oceanographic campaigns. In particular, we want to thank: the crews from the CNRS/INSU and Genavir vessels, Pierre Guéguéniat, Dominique Boust, Rémi Gandon, René Léon, Marianne Lamotte, Bruno Fiévet, Olivier Connan, Pascal Morin, Bernard Le Cann, Louis Marié, Robert Lafite, Kins Leonard, Peter Kershaw and Jürgen Hermann. The Orano La Hague SPR team must also be thanked for providing detailed liquid radioactive releases during and after oceanographic campaigns. Lastly, we thank the editors and anonymous reviewers for their fruitful comments during the review process.

**Conflicts of Interest:** The authors certify that they have no conflicts of interest to declare.

# Appendix A

**Table A1.** List of oceanographic campaigns accounted in the database. Positions are referenced in WGS84 in decimal degrees (N, E: + ; S, W:-). Radionuclide concentrations are in Bq·m$^{-3}$. Abbreviations: LH: La Hague Cape; EC:  English Channel; CI: Channel Islands; CS: Celtic Sea; IS: Irish Sea; NS: North Sea;  MS: Mediterranean Sea; BB: Biscay Bay;  NA: North Atlantic.

| Campaign name | Year | Area abr. | Stat. nb. | Lon. Min. | Lon. Max. | Lat. min. | Lat. max. | H3 | | Cs137 | | Cs134 | | Sb125 | | Ru106 | | Co60 | |
|---|---|---|---|---|---|---|---|---|---|---|---|---|---|---|---|---|---|---|---|
| | | | | | | | | Nb. | Max. meas. | Nb. | Max. meas. | Nb. | Max. meas. | Nb. | Max. meas. | Nb. | Max. meas. | Nb. | Max. meas. |
| Cirolana0582 | 1982 | EC, CS, NS | 42 | -7.0 | 2.0 | 49.5 | 58.7 | 39 | **642** | | | | | | | | | | |
| Cirolana0583 | 1983 | EC, CS, IS | 54 | -7.5 | -1.9 | 49.3 | 54.7 | 44 | **13283** | 22 | **688** | | | 14 | **459** | 36 | **7511** | 1 | **21.8** |
| Clione0983 | 1983 | EC, CS | 36 | -8.5 | 1.3 | 48.6 | 51.6 | 35 | **37.0** | | | | | | | | | | |
| GEA0983 | 1983 | EC, CI | 12 | -2.5 | -2.2 | 49.3 | 49.6 | 12 | **19.2** | 6 | **3.0** | | | 9 | **23.8** | 9 | **69** | | |
| Thalassa1283 | 1983 | EC, BB | 56 | -4.8 | 0.3 | 47.4 | 50.0 | 55 | **59.2** | 31 | **10.0** | | | 45 | **149** | 44 | **207** | 8 | **44.4** |
| Pluteus0583 | 1983 | EC, CI, NS | 119 | -2.7 | 2.9 | 48.9 | 51.3 | 115 | **69.2** | 21 | **15.5** | | | 105 | **1668** | 79 | **3256** | | |
| Pluteus0686 | 1986 | EC, CI | 185 | -4.8 | 2.0 | 48.3 | 51.3 | 184 | **34.5** | 141 | **6.7** | | | 163 | **211** | 134 | **686** | 29 | **13.0** |
| Ferry1986 | 1986 | EC, CS | 31 | -8.3 | -4.0 | 48.8 | 51.8 | 31 | **15.4** | 7 | **5.3** | | | | | | | | |
| Pluteus1286 | 1986 | EC | 47 | -1.8 | -1.4 | 49.7 | 50.3 | 47 | **18.6** | 27 | **5.2** | | | 44 | **89.7** | 38 | **180** | 27 | **8.8** |
| GEA1986 | 1986 | EC, CS | 97 | -6.0 | 1.9 | 48.7 | 51.2 | 97 | **53.3** | 80 | **21.8** | | | 45 | **46.3** | 34 | **315** | 13 | **8.9** |
| Nies0187 | 1987 | NS | 29 | 1.5 | 9.2 | 51.0 | 55.0 | 29 | **84.0** | 23 | **13.0** | | | | | | | | |
| Aurélia0387 | 1987 | NS | 31 | 3.0 | 4.7 | 51.7 | 53.0 | 31 | **15.5** | 31 | **5.7** | | | 30 | **57.1** | 29 | **213** | | |
| Sépia1987 | 1987 | EC, NS | 14 | 1.3 | 2.3 | 50.3 | 51.2 | 14 | **11.2** | 4 | **1.8** | | | 14 | **48.5** | 14 | **121** | 1 | **0.4** |
| Manche0687 | 1987 | EC | 50 | -1.9 | 0.3 | 49.6 | 50.8 | 50 | **10.0** | 10 | **3.0** | | | 50 | **84.1** | 40 | **495** | 7 | **10.4** |
| Luctor0687 | 1987 | NS | 63 | 2.0 | 3.9 | 51.1 | 51.9 | 63 | **31.2** | 19 | **4.0** | | | 61 | **62.2** | 19 | **120** | | |
| Pluteus0887 | 1987 | EC, CI, NS | 165 | -4.0 | 7.9 | 48.7 | 54.2 | 164 | **90.5** | 84 | **39.6** | | | 152 | **70.1** | 139 | **180** | 14 | **2.5** |
| Aurélia1187 | 1987 | NS | 6 | 3.5 | 4.6 | 52.1 | 52.7 | 6 | **12.8** | 4 | **1.6** | | | 4 | **57.4** | 6 | **104** | | |
| GEA 1987 | 1987 | EC | 16 | -2.0 | 2.5 | 49.5 | 51.2 | 16 | **11.8** | 11 | **2.6** | | | 16 | **44.8** | 13 | **143** | 5 | **2.2** |
| Aurélia0488 | 1988 | NS | 9 | 3.5 | 4.6 | 52.1 | 52.9 | 9 | **12.6** | | | | | 6 | **58.0** | | | | |
| Pluteus0488 | 1988 | EC | 44 | -1.7 | 1.6 | 49.5 | 51.1 | 44 | **11.1** | 1 | **1.1** | | | 43 | **43.6** | 28 | **116** | | |
| Pluteus0688 | 1988 | EC | 105 | -5.1 | 1.6 | 49.4 | 51.1 | 104 | **12.2** | | | | | 81 | **112** | 63 | **222** | | |
| Pluteus0788 | 1988 | EC, CI | 103 | -4.0 | -1.1 | 48.6 | 49.8 | 103 | **12.5** | 4 | **1.4** | | | 94 | **464** | 42 | **324** | 6 | **2.7** |
| GEA0788 | 1988 | EC | 105 | -5.0 | 1.6 | 49.4 | 51.1 | 104 | **12.2** | 21 | **1.9** | | | 81 | **180** | 64 | **222** | 5 | **2.6** |
| Tramanor1 | 1988 | EC, NS | 256 | -2.0 | 9.0 | 49.7 | 60.2 | 242 | **70.7** | 192 | **13.3** | | | 186 | **48.9** | 107 | **189** | 2 | **2.0** |
| Gedynor | 1989 | EC, NS | 127 | -4.0 | 8.5 | 49.2 | 58.9 | 119 | **38.8** | 37 | **4.3** | | | 89 | **57.1** | 33 | **151** | 1 | **1.6** |
| RadeBrest0689 | 1989 | CI | 23 | -4.6 | -4.3 | 48.3 | 48.4 | 23 | **5.9** | | | | | | | | | | |
| RadeBrest0789 | 1989 | CI | 23 | -4.6 | -4.3 | 48.3 | 48.4 | 23 | **6.3** | | | | | | | | | | |
| GEA0989 | 1989 | EC, CI | 88 | -2.6 | -0.9 | 49.4 | 50.2 | 87 | **31.5** | 57 | **7.8** | | | 88 | **107.4** | 80 | **355** | 23 | **15.9** |
| Pluteus0989 | 1989 | EC, CI | 124 | -2.8 | -1.0 | 49.2 | 50.3 | 124 | **37.0** | 71 | **4.8** | | | 123 | **203.0** | 100 | **985** | 53 | **13.9** |

| | | | | | | | | | | | | | | | | | | | |
|---|---|---|---|---|---|---|---|---|---|---|---|---|---|---|---|---|---|---|---|
| **RadeBrest0390** | 1990 | CI | 46 | -4.6 | -4.2 | 48.3 | 48.4 | | | 46 | **4.1** | | | | | | | | |
| **Tramanor2** | 1990 | EC, CI, NS | 205 | -6.0 | 8.5 | 48.3 | 60.1 | | | 171 | **23.7** | 59 | **2.6** | 137 | **14.5** | 116 | **90** | 39 | **15.2** |
| **FluxManche0990** | 1990 | EC | 32 | 1.2 | 1.6 | 50.7 | 51.1 | | | 24 | **14.8** | 9 | **57.0** | 26 | **23.1** | 17 | **56** | | |
| **Cirolana1** | 1991 | EC, NS | 55 | -1.7 | 4.5 | 49.8 | 55.2 | | | 44 | **21.9** | 11 | **1.1** | 35 | **13.8** | 33 | **67** | 24 | **2.7** |
| **Tramanor3** | 1991 | EC, NS | 162 | -1.8 | 9.1 | 49.4 | 63.3 | | | 144 | **28.0** | 65 | **1.7** | 116 | **18.9** | 73 | **37** | 41 | **1.4** |
| **Cirolana2** | 1992 | EC, NS | 42 | -1.1 | 5.6 | 49.7 | 55.2 | | | 38 | **18.5** | 3 | **0.5** | 29 | **11.4** | 18 | **8** | 11 | **3.3** |
| **GolfeNormand Breton0492** | 1992 | EC, CI | 39 | -3.0 | -1.7 | 48.7 | 49.8 | | | 37 | **7.5** | 24 | **0.5** | 37 | **21.0** | 7 | **14** | 15 | **2.4** |
| **Valdivia1993** | 1993 | EC, CS, IS, NS | 63 | -6.0 | 11.0 | 49.8 | 58.5 | | | 62 | **259** | 60 | **2.1** | 60 | **13.8** | 58 | **10** | 56 | **3.9** |
| **Arrho1994** | 1994 | MS | 9 | 4.7 | 4.9 | 43.3 | 43.4 | | | 9 | **5.1** | 5 | **1.1** | 5 | **3.6** | 7 | **68** | 5 | **0.4** |
| **Gedymac1994** | 1994 | EC, CI, CS, IS | 222 | -8.4 | 1.5 | 48.0 | 54.5 | 95 | **8870** | 211 | **222** | 55 | **1.4** | 95 | **30.8** | 33 | **9** | 43 | **2.5** |
| **Dymanche1994** | 1994 | EC | 81 | -1.9 | 1.5 | 49.3 | 51.1 | | | 81 | **27.7** | 45 | **2.4** | 47 | **5.8** | 6 | **13** | 18 | **1.3** |
| **RadialeCh-Wh0295** | 1995 | EC | 8 | -1.7 | -1.0 | 49.7 | 50.7 | 4 | **5860** | 8 | **5.6** | | | 1 | **2.4** | 2 | **5** | 5 | **0.9** |
| **RadialeCh-Wh0395** | 1995 | EC | 7 | -1.7 | -1.0 | 49.7 | 50.7 | | | 6 | **4.4** | | | 1 | **1.5** | | | 2 | **0.5** |
| **RadialeCh-Wh0595** | 1995 | EC | 8 | -1.7 | -1.0 | 49.7 | 50.7 | | | 8 | **4.7** | 1 | **0.3** | 3 | **1.3** | 1 | **4** | 3 | **0.5** |
| **Omex 0695** | 1995 | CI, NA | 7 | -13.7 | -6.6 | 47.4 | 56.6 | | | 7 | **1.7** | | | | | | | | |
| **Omex 0895** | 1995 | CI, NA | 8 | -14.1 | -7.4 | 47.7 | 50.6 | | | 8 | **3.3** | | | | | | | | |
| **RadialeCh-Wh0795** | 1995 | EC, CI | 31 | -2.7 | -1.0 | 49.3 | 50.6 | | | 31 | **9.0** | 25 | **0.7** | 28 | **1.6** | 21 | **2** | 25 | **0.9** |
| **RadialeCh-Wh0995** | 1995 | EC | 8 | -1.9 | -1.1 | 49.7 | 50.6 | | | 8 | **7.2** | 4 | **0.3** | 5 | **3.6** | 4 | **10** | 8 | **0.6** |
| **RadialeCh-Wh1195** | 1995 | EC, CI | 37 | -2.6 | -1.6 | 49.0 | 49.9 | | | 37 | **5.8** | 21 | **0.2** | 35 | **2.0** | 33 | **34** | 28 | **0.7** |
| **Ferry0396** | 1996 | EC, CS | 23 | -8.3 | -4.0 | 48.7 | 51.8 | 4 | **719** | 23 | **2.7** | | | | | | | 3 | **0.7** |
| **FondDeBaie0496** | 1996 | EC | 15 | -2.3 | -1.6 | 48.6 | 48.8 | 5 | **3710** | 14 | **3.4** | 1 | **0.1** | 13 | **0.7** | 3 | **1** | 9 | **0.6** |
| **Ferry0696** | 1996 | EC | 8 | -1.9 | -1.7 | 49.7 | 50.7 | | | 8 | **5.1** | 2 | **0.5** | 5 | **2.8** | 4 | **3** | 6 | **0.5** |
| **FondDeBaie0796** | 1996 | EC | 10 | -2.4 | -2.0 | 48.6 | 48.7 | 10 | **3200** | 10 | **3.3** | | | 7 | **0.7** | | | 4 | **0.4** |
| **Irma1996** | 1996 | EC | 83 | -6.0 | -1.3 | 48.1 | 50.5 | 60 | **21000** | 71 | **6.1** | 1 | **0.3** | 9 | **2.1** | 10 | **11** | 19 | **6.1** |
| **Arcane1997** | 1997 | CS, NA | 52 | -12.8 | -3.3 | 39.1 | 45.6 | 22 | **268** | 43 | **2.5** | | | | | | | | |
| **Ferry1997** | 1997 | EC | 19 | -4.2 | -1.6 | 48.8 | 50.7 | 11 | **484** | 19 | **4.2** | | | | | | | 1 | **0.9** |
| **FluxSed1998** | 1998 | EC, CI | 66 | -6.3 | 0.1 | 48.6 | 50.2 | | | 66 | **5.9** | | | 5 | **1.9** | 9 | **13** | 8 | **1.0** |
| **Atmara1998** | 1998 | EC, CS, NA | 196 | -14.0 | -3.6 | 45.5 | 55.7 | 130 | **1930** | 191 | **16.1** | | | | | | | | |
| **Cirolana2000** | 2000 | CS, BB | 35 | -11.5 | 1.5 | 47.3 | 52.3 | 35 | **1323** | | | | | | | | | | |
| **Ovide2002** | 2002 | NA | 60 | -42.6 | -6.2 | 35.8 | 59.8 | 45 | **377** | 15 | **2.7** | | | | | | | | |
| **Dispro08** | 2002 | LH, EC, CI | 1029 | -2.4 | 0.9 | 49.4 | 50.0 | 845 | **45103** | | | | | | | | | | |
| **Dispro09** | 2002 | LH, EC, CI | 1058 | -2.4 | -1.6 | 49.5 | 49.8 | 1046 | **2355208** | | | | | | | | | | |
| **Dispro10** | 2002 | LH, EC, CI | 1899 | -2.2 | -1.6 | 49.6 | 49.8 | 1890 | **3604897** | | | | | | | | | | |
| **Dispro11** | 2003 | LH, EC, CI | 2398 | -2.1 | -1.7 | 49.5 | 49.8 | 2397 | **1532137** | | | | | | | | | | |
| **Dispro12** | 2003 | LH, EC, CI | 640 | -2.9 | -1.7 | 48.7 | 49.9 | 587 | **12111** | | | | | | | | | | |
| **Ovide2004** | 2004 | NA | 31 | -42.9 | -9.8 | 40.3 | 59.9 | 17 | **236** | 14 | **2.4** | | | | | | | | |
| **Dispro2004** | 2004 | LH, EC, CI | 3539 | -2.9 | -1.2 | 48.6 | 50.0 | 3537 | **392028** | | | | | | | | | | |
| **Dispro2005** | 2005 | LH, EC, CI | 4143 | -2.4 | 1.5 | 49.3 | 50.7 | 4103 | **268816** | | | | | | | | | | |

| | | | | | | | | | | | | | | | | | | | |
|---|---|---|---|---|---|---|---|---|---|---|---|---|---|---|---|---|---|---|---|
| **Disver2008** | 2008 | LH, EC, CI | 796 | -2.0 | -2.0 | 49.7 | 49.7 | 784 | **190660** | | | | | | | | | | |
| **Aspex2009** | 2009 | BB | 36 | -6.0 | -1.5 | 44.0 | 47.8 | 34 | **336** | | | | | | | | | | |
| **Ovide2010** | 2010 | NA | 1 | -19.1 | -19.1 | 45.8 | 45.8 | 1 | **76** | | | | | | | | | | |
| **Aspex2010** | 2010 | BB | 62 | -6.4 | -1.6 | 44.0 | 48.3 | 62 | **266** | | | | | | | | | | |
| **Disver2010** | 2010 | LH | 5556 | -2.0 | -2.0 | 49.7 | 49.7 | 5314 | **7029776** | | | | | | | | | | |
| **Disver2011** | 2011 | LH | 12498 | -2.0 | -1.9 | 49.7 | 49.8 | 12498 | **8739130** | | | | | | | | | | |
| **Aspex2011** | 2011 | BB | 74 | -6.0 | -1.5 | 44.0 | 47.8 | 62 | **211** | | | | | | | | | | |
| **Trimadu2013** | 2013 | NA | 18 | -13.2 | 55.5 | -32.4 | 34.9 | 17 | **80** | 12 | **1.1** | | | | | | | | |
| **Traces2014** | 2014 | EC, CI | 1205 | -3.0 | -1.6 | 48.6 | 49.8 | 1170 | **125159** | | | | | | | | | | |
| **Traces2015** | 2015 | EC, CI | 547 | -3.0 | -1.7 | 48.8 | 49.8 | 546 | **38320** | | | | | | | | | | |
| **Dynsedim2016** | 2016 | BB | 31 | -5.3 | -2.6 | 45.6 | 47.9 | 30 | **640** | | | | | | | | | | |
| **Pelgas2016** | 2016 | BB | 130 | -5.8 | -1.3 | 43.7 | 47.9 | 125 | **583** | | | | | | | | | | |
| **Plume2016** | 2016 | BB | 254 | -5.2 | -1.1 | 44.7 | 48.6 | 177 | **19072** | 12 | **1.3** | | | | | | | | |
| **Goury (time series)** | 1984-2018 | LH, EC | 744 | -1.9 | -1.9 | 49.7 | 49.7 | 266 | **41999** | 615 | **79.9** | 260 | **62.3** | 498 | 223.1 | 484 | **1745** | 398 | **50.7** |
| **Nb. total (campaigns):** | | | 39642 | | | | | 35663 | | 3492 | | 1295 | | 2242 | | 1606 | | 568 | |
| **Nb. Total (all):** | 80 | | 40386 | | | | | 35929 | | 4107 | | 1555 | | 2740 | | 2090 | | 966 | |

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
