# Peer review of "Dissolved Radiotracers and Numerical Modeling in North European Continental Shelf Dispersion Studies (1982–2016): Databases, Methods and Applications"

_water, doi:10.3390/w12061667_

Round 1

Reviewer 1 Report

This article presents an extensive review of artificial dissolved radionuclides distribution in coastal seas, with a focus on the European coastal shelfs. This database is of high importance regarding geochemistry but also to validate dispersion models at different spatial and temporal scales. It is a very well documented article. There is no doubt on its interest.

I would have few suggestions:

- the title needs to be improved and in particular the word “artificial” is required

- a table with the different radionuclides and their half-life is required to help the reader

- lines 33-38: there is also the Chernobyl  accident

-the article still sounds like a report. The authors need to reduce the references to the IRSN- -LRC. Although this lab did and does a great job, the article must be more neutral. It is an article on artificial radionuclides and not on this lab. Most references to this lab could be replaced by references.

The introduction is too long, and needs to be shortened. It is a very long article, it is necessary that the authors of the article more directly to the essential. Although correctly written, english polishing could also improve the reading comfort.

Author Response

Answers to Reviewer 1

Comments and Suggestions for Authors

I would have few suggestions:

- the title needs to be improved and in particular the word “artificial” is required

Done

- a table with the different radionuclides and their half-life is required to help the reader

Done in section 2.1.: Table 1

- lines 33-38: there is also the Chernobyl  accident

Done

-the article still sounds like a report. The authors need to reduce the references to the IRSN- -LRC. Although this lab did and does a great job, the article must be more neutral. It is an article on artificial radionuclides and not on this lab. Most references to this lab could be replaced by references.

Done

The introduction is too long, and needs to be shortened. It is a very long article, it is necessary that the authors of the article more directly to the essential.

As it is a review paper, it appears important to describe the general context and previous studies that are not well known in this journal. Re-organisation was done to reduce methodology description, but a minimum is requested to understand the results

Although correctly written, english polishing could also improve the reading comfort.

Done by a professional translator

Reviewer 2 Report

This paper presents the possibility to apply the radionuclide measurement and release fluxes to validate dispersion hydrodynamic models. This is an interesting purpose and the focus on hydrodynamic modelling in short/large temporal and spatial scales is consistent with the scopes of the journal. However, I think that the paper is too dispersive because it describes in detail the aspects related to the sampling and to the characteristics of the radionuclides, i.e. pages from 4 to 8. On the other hand, the description concerning numerical modelling is very limited and is not sufficient to make the reader understand how it was applied. Moreover, it seems that the novelty regards the application of the hydrodynamic model at different scales to reproduce in realistic conditions the measured dispersion. Indeed, this has already been done and published by the authors themselves, but in the introduction this aspect is not well declared. In fact, lines 62-69 and 87-91 have no references about numerical modelling and approaches. Also in the following, it is not completely clear what is new and what has been already study also from a numerical point of view.

In this sense, in the document available online titled “Dispersion des radionucléides dans les mers du nord-ouest de l’Europe: observations et modélisation” by Pascal Bailly Du Bois (November 2014), I have found the same figures included in the present paper, regarding also the comparison between modelled and measured results. I refer for example to Figures 1,2,4,6,8,9,13,14,15,16 and the data in Table 2. They are identical to those presented in the previous report. So, what is the novelty respect to a work done 5 years ago if the results are the same?

I don’t understand at this point the purpose of the authors since they have omitted or not clearly pointed out these aspects thorough the paper.

Author Response

Answers to Reviewer 2

Comments and Suggestions for Authors

However, I think that the paper is too dispersive because it describes in detail the aspects related to the sampling and to the characteristics of the radionuclides, i.e. pages from 4 to 8. On the other hand, the description concerning numerical modelling is very limited and is not sufficient to make the reader understand how it was applied. Moreover, it seems that the novelty regards the application of the hydrodynamic model at different scales to reproduce in realistic conditions the measured dispersion. Indeed, this has already been done and published by the authors themselves, but in the introduction this aspect is not well declared. In fact, lines 62-69 and 87-91 have no references about numerical modelling and approaches. Also in the following, it is not completely clear what is new and what has been already study also from a numerical point of view.

In this sense, in the document available online titled “Dispersion des radionucléides dans les mers du nord-ouest de l’Europe: observations et modélisation” by Pascal Bailly Du Bois (November 2014), I have found the same figures included in the present paper, regarding also the comparison between modelled and measured results. I refer for example to Figures 1,2,4,6,8,9,13,14,15,16 and the data in Table 2. They are identical to those presented in the previous report. So, what is the novelty respect to a work done 5 years ago if the results are the same?

I don’t understand at this point the purpose of the authors since they have omitted or not clearly pointed out these aspects thorough the paper.

It was stated at the beginning that the paper is a synthesis of previous works. The paper proposed to the special issue was in this context.

To be clearer, the paper will be presented as a review paper.

Nevertheless, most of the data attached as supplementary material in the PANGAEA database were not available before (64 % of radionuclide measurements and 93% of releases data), and some results were not easily accessible before. The purpose of this paper is to provide a comprehensive access to the existing data, methods and results.

For example, the documents “Dispersion des radionucléides dans les mers du nord-ouest de l’Europe: observations et modélisation” and "Mesure et modélisation de la dispersion verticale dans le raz Blanchard " are in French and difficult to read by most of the scientific community. The old results are the same but some recent ones where not described (3D dispersion at short scale and in the Biscay Bay).

Round 2

Reviewer 1 Report

The authors have considered seriously the comments of the reviewers. THey did a good job to improve the manuscript. 

This work is an excellent review article that desserves publications, and even more as it présents also unpublisshed results.

Author Response

Thanks a lot for your evaluation

Reviewer 2 Report

In the first version of the article it was not so clear what was new and what was already published. Furthermore, many figures were copied identical from other works previously published by the same authors. This is plagiarism. For this reason, in the current version, the authors have changed the type of the manuscript, from article to review. A review provides a summary of the state of the art on a particular topic. This is in fact a synthesis, but of the work done by the authors themselves. There are 87 references, 41 of them are own papers of the authors. This proportion is really unbalanced. More than a review it looks like an autobiography ...

If there is really new data that has not been previously published, then the authors can rethink their work by focusing only on what has been done again, or if they want to make a review they must consider the contributions of other authors on the same subject, with a fair proportion between their own work and that of others. There are many figures and data identical (and not redrawn...see the first review report) to those previously published but the permission of the publishers (for example Elsevier) is not indicated. So in my opinion there is still plagiarism and I think the work should not be published.

Author Response

The principal objective of this work is to make accessible a comprehensive database to perform accurate model – measurement comparisons at the scale of the Northern European continental shelf. This database would be unfruitful if it is not explained why and how it was built and applied. This justifies the presentation of methods and results already published and give ways for future works using these data and methods. Some results are completely new or inaccessible in open literature.

The word plagiarism used is unfounded and unacceptable. All references are given, me and co-authors have right to present and explain our own works and all figures was re-edited for this publication.